

# Effects of local advection on the spatial sensible heat flux variation on a mountain glacier

Tobias Sauter[1,2] and Stephan P. Galos[2]

[1]Climate System Research Group, Institute of Geography, Friedrich-Alexander University Erlangen-Nürnberg (FAU), Erlangen, Germany.
[2]Institute of Atmospheric and Cryospheric Sciences, University of Innsbruck, Austria.

*Correspondence to:* Tobias Sauter (tobias.sauter@fau.de)

**Abstract.**

Distributed mass balance models, which translate micrometeorological conditions into local melt rates, have proven deficient to reflect the energy flux variability on mountain glaciers. This deficiency is predominantly related to shortcomings in the representation of local processes in the forcing data. We found by means of idealized Large-Eddy Simulations that heat

advection, associated with local wind systems, cause small-scale heat flux variations by up to $100\,\mathrm{Wm^{-2}}$ during clear sky conditions. Here we show that process understanding at a few on-glacier sites is insufficient to infer on the wind and temperature distributions across the glacier. On average, glacier heat fluxes are both over- and underestimated by up to $16\,\mathrm{Wm^{-2}}$ when using extrapolated temperature and wind fields. The sign and magnitude of the errors depend on the site selection as well as on the flow direction. Our results demonstrate how the shortcomings in the local heat flux estimates are related to topographic

effects and the insufficient characterisation of the temperature advection process. The magnitudes of the surface heat flux errors are strong enough to significantly affect the surface energy balance and derived climate sensitivities of mountain glaciers.

## 1 Introduction

The complex interaction of glaciers, atmosphere and topography constitutes a fundamental challenge in glaciological research. Countless studies aim to identify the climatic drivers behind observed glacier changes by using distributed mass and energy

balance models (e.g. Arnold et al., 1996; Hock and Holmgren, 2005; Klok and Oerlemans, 2002; Mölg et al., 2009). While these kind of models summarize our understanding of the governing physical processes at a point scale, they have proven deficient to reflect the variability of the energy and mass fluxes on mountain glaciers (e.g. Gurgiser et al., 2013; MacDougall and Flowers, 2011; Prinz et al., 2016). The loss of information in space and time predominantly results from shortcomings in the forcing data, which in turn expresses the need for better representation of the local processes.

While large-scale weather shapes the environmental conditions in which mountain glaciers exist, the mass and energy exchange on individual glaciers is controlled by the micrometeorological conditions. Given the complex topography around mountain glaciers with its contrasting surface characteristics, it is not trivial to bridge the scale gap between the large-scale conditions and the local characteristics. The micrometeorological condition of the surface layer is directly influenced by the presence of the earth's surface and quickly responds to changes in the surface energy budget. The radiative and turbulent heat



fluxes cool and heat the near-surface air layer and determine the temperature distribution across the topography. Local temperature excess and deficit create buoyancy forces that drive the thermal wind systems, including the valley circulations, slope and glacier winds (e.g. Munro and Scott, 1989; Oerlemans and Grisogono, 2002; Sicart et al., 2014; Smeets et al., 2000; van den Broeke, 1997). The thermal wind phenomena are often superimposed and partly overwhelmed by the dynamically-driven winds, which in turn are characterised by topographic effects. Heat advection associated with the mean flow and intermittent turbulent mixing events alter the thermal conditions and, finally, link it to the large-scale weather.

The fluctuations of the thermal conditions are of practical interest for glacier mass balance studies. For example, winds may advect warm air from the surroundings towards the glacier which locally increase the downward directed sensible heat flux (Ayala et al., 2015; Shea and Moore, 2010; Hannah et al., 2000; Moore and Owens, 1984; Strasser et al., 2004). The energy surplus can be critical for the ablation, considering that the turbulent heat flux can make over 50% of the total energy during large melt events on mountain glaciers in summer (e.g. Cullen and Conway, 2015; Gillett and Cullen, 2011; Van den Broeke, 1997; Hock, 2005; Klok and Oerlemans, 2002; Oerlemans and Klok, 2002; Giessen et al., 2008; Moore and Owens, 1984). Therefore, whenever small scale variations of the melt rates are of peculiar interest, a profound knowledge of the advection processes and the micrometeorological characteristics is needed.

A fundamental obstacle in studying small-scale boundary layer characteristics is, that even on well-studied mountain glaciers, the deficiency of monitoring activities restricts the process understanding, required for detailed research, to a few sites and limited time periods (e.g. Wagnon et al., 1999; Mölg and Hardy, 2004; Obleitner and Lehning, 2004; Reijmer and Hock, 2008; Nicholson et al., 2013). The phenomenological knowledge that is valid for the specific location and weather situation does not have greater significance beyond the case (e.g. Machguth et al., 2006; Gardner et al., 2013; Zemp et al., 2013). This constraint makes it challenging to infer on micrometeorological conditions from a limited number of observations. Glaciological modelling studies typically circumvent this obstacle by constructing meteorological forcing fields, e.g. for temperature and wind, from scattered observations using fixed or variable lapse rates (e.g. Greuell and Böhm, 1998; Carturan et al., 2015; Ayala et al., 2015; Petersen et al., 2013; Huintjes et al., 2015; Weidemann et al., 2013; Jarosch et al., 2012). The interpolated fields then serve for the estimation of turbulent fluxes at any given point on the glacier. As a result of the simplified assumptions, the modelled sensible heat flux distribution is unlikely to truly reflect the full variability in time and space. It is still an open scientific issue how these assumptions impact the estimated local and glacier-wide melting rates. However, this question cannot be answered by means of a few individual observations.

To overcome this difficulty, we make use of high resolution Large-Eddy Simulations (LES). The LES are considered as pseudo-reality - a testbed to identify the shortcomings in the local surface heat flux estimates when the lack of observations restrict our micrometeorological knowledge to a few sites. The pseudo-reality atmosphere is not required to be an observed real world case, but needs to be plausible in the sense that relevant processes are realistically simulated.

After a brief description of the LES model (Section 2), we show that the pseudo-reality realistically describes the relevant atmospheric processes observed in a glaciated mountainous region (Section 3). We begin by exploring the mean flow fields, the turbulence characteristics and then address the spatial variations of the surface sensible heat flux. In Section 4, we use the



bulk approach in concert with linearly interpolated fields based on virtual sites and analyse the impacts on the variability of the surface sensible heat flux. The last section provides a summary of the main findings.

## 2 Methodology

### 2.1 Large-Eddy Simulation solver

The pseudo-reality atmosphere is simulated by an OpenFOAM-based incompressible LES solver (Churchfield et al., 2014). The solver is based on the incompressible filtered Navier-Stokes equations, using the Boussinesq approximation for buoyancy, along with the continuity equation

$$\frac{\partial \overline{U}_i}{\partial x_i} = 0. \tag{1}$$

The filtered momentum equation is given as

$$\frac{\partial \overline{U}_j}{\partial t} + \frac{\partial \overline{U}_i \overline{U}_j}{\partial x_i} = -2\epsilon_{i3k}\Omega \overline{U}_k - \frac{\partial \overline{p}}{\partial x_j} - \frac{\tau_{ij}{}^r}{\partial x_i} - \rho_b g_j, \tag{2}$$

with the overline denoting the LES filtering operation, $\overline{U}_i$ the component of the resolved-scale velocity vector in the direction $x_i$, $\epsilon_{ijk}$ the alternating unit tensor, $\Omega$ the planetary rotation rate vector, $\overline{p}$ the pressure, and $g_i$ the gravitation vector. The strength and the sign of the buoyancy force $\rho_b$ is given by

$$\rho_b = 1 - \left(\frac{\overline{\theta} - \theta_0}{\theta_0}\right), \tag{3}$$

where $\overline{\theta}$ is the resolved-scale potential temperature and $\theta_0$ is a reference temperature. In practise the isotropic part (residual kinetic energy, $k_r \equiv 0.5\,\tau_{ij}^R$) of the residual-stress tensor $\tau_{ij}^R = \overline{U_i U_j} - \overline{U}_i\,\overline{U}_j$ is absorbed into the filtered pressure term, and only the anisotropic residual-stress tensor

$$\tau_{ij}{}^r \equiv \tau_{ij}{}^R - \frac{2}{3}k_r\delta_{ij}, \tag{4}$$

need to be modelled (also called subgrid-scale stress tensor). As the vast majority of LES studies for SBL, we use a dynamic 20 Smagorinsky SGS model which relies on the eddy-viscosity assumption to close Equation 2. The model relates the residual stresses to the resolved large-scale velocity deformation

$$\tau_{ij}{}^r = -2\nu_t \overline{S}_{ij}, \tag{5}$$



where $\nu_t$ is the eddy-viscosity of the residual motions, and

$$\overline{\boldsymbol{S}}_{ij} = \frac{1}{2}\left(\frac{\partial \overline{\boldsymbol{U}}_i}{\partial \boldsymbol{x}_j} + \frac{\partial \overline{\boldsymbol{U}}_j}{\partial \boldsymbol{x}_i}\right) \tag{6}$$

the resolved-scale strain rate tensor. The eddy-viscosity

$$\nu_t = {\ell_s}^2 \overline{\mathcal{S}} = (C_s \overline{\Delta})^2 \overline{\mathcal{S}}$$

is taken to be proportional to the Smagorinsky lengthscale, $\ell_s$, and the characteristic filtered rate of strain $\overline{\mathcal{S}} = (2\overline{\boldsymbol{S}}_{ij}\overline{\boldsymbol{S}}_{ij})^{1/2}$. The lengthscale is usually modelled by a fixed constant $C_s$ and the filter width $\overline{\Delta}$. At high Reynolds-number turbulence, with $\overline{\Delta}$ in the inertial subrange, the resolved scales account for nearly all of the kinetic energy (Pope, 2000). According to the model, the energy transfer from the resolved-scale eddies to the residual motions is entirely balanced by the dissipation of kinetic energy (Churchfield et al., 2014). While, in the mean, energy is transferred from the large to small scales, is has

been recognized that locally there can be significant backscatter of energy from the residual motions on the resolved scales (Pope, 2000). Furthermore, Equation 5 is only valid for isotropic turbulence and is therefore not strictly applicable to complex terrain. The importance of the effects of backscatter and anisotropy for SBL has been shown by Kosovic and Curry (2000). Nevertheless, we assume that the details of the model are of minor importance and the effect of anisotropy becomes negligible when the grid-scale is small compared to the energy containing turbulent scales.

In the original formulation the value of the constant $C_s = 0.17$ is derived from the Kolmogorov spectrum assuming that the transfer of energy to the residual motions is balanced by the dissipation. This constant, however, is not ideal for all locations of the flow (Churchfield et al., 2014), i.e. in regions where the buoyancy flux extinguishes the turbulence the residual shear stresses should be zero. In general, the value of $C_s$ should become zero in the limit of laminar flow and any non-zero value of the coefficient would incorrectly lead to residual shear stresses. To overcome this issue Meneveau et al. (1996) proposed

a Lagrangian-averaged dynamic Smagorinsky model that allows the coefficient to vary in time and space based on the flow (Anderson and Meneveau, 1999; Sarghini et al., 1999; Bou-Zeid et al., 2005). This type of closure is appropriate for flow over complex terrain and is therefore used in this study (Bou-Zeid et al., 2005).

     Proceeding from the instantaneous internal energy equation the conservation of potential temperature can be derived, and becomes

$$\frac{\partial \overline{\theta}}{\partial t} + \frac{\partial \overline{\boldsymbol{U}}_i \overline{\theta}}{\partial \boldsymbol{x}_i} = -\frac{\boldsymbol{\tau}_{\theta i}}{\partial \boldsymbol{x}_j}, \tag{7}$$

where $\boldsymbol{\tau}_{\theta i}$ is the SGS temperature flux given by

$$\boldsymbol{\tau}_{\theta i} = -\frac{\nu_t}{Pr_t}\frac{\partial \overline{\theta}}{\partial x_i}, \tag{8}$$





where $Pr_t$ is the turbulent Prandtl number. Changes in temperature by radiative forcing and phase change of water are neglected in this study.

The filtered momentum equation is solved using the PIMPLE algorithm, and a preconditioned bi-conjugate gradient solver for asymmetric matrices. To reduce numerical dissipation the convective terms are solved using a second-order central dif-
ferencing scheme with a multi-dimensional limiter. The time derivative is discretized by a second-order implicit scheme with adaptive time stepping.

## 2.2 Study Area

Even though the LES is designed as pseudo-reality, the lower boundary condition is provided by a real topography. The designated study area is located at the head of Martell Valley in the central Ortler-Cevedale Group, Autonomous Province
of Bozen, Northern Italy ($46.28°$ N, $10.60°$ E; see Figure 1). The model domain comprises a major part of the contiguous glaciated area covering the northern section of the Cevedale Massif, the summit of which is the highest point of the study area (3769 m a.s.l.). Three glaciers connected to each other are in the focus of the study: Fürkele Ferner, Zufallferner, and Langenferner. The surface area of the glaciers is about 6.62 $km^2$ (2013) with an altitudinal extent from about 3750 m a.s.l near the summit of Hintere Zufallspitze, down to 2595 m a.s.l. at the lowest point of Zufallferner. The model domain includes a
wide variety of topographic features such as steep slopes, glaciated and unglaciated (summit-) ridges of various aspects, as well as larger glacier sections with smooth terrain and low slope angles. The topography can be regarded as (i) representative for many glaciers in the European Alps, and (ii) highly suitable for investigating the complex interaction of large-scale (synoptic) forcing and small-scale topographic features.

## 2.3 Initial Conditions

The surface temperature, $T_s$, of both the glacier surface and the surrounding topography is uniformly initialized with 273.16 K. The atmospheric background state for temperature and pressure is derived from ERA-Interim data. To avoid temperature jumps between the free atmosphere and the underlying surface an analytical Prandtl model for thermally induced slope flows is applied as proposed by Oerlemans and Grisogono (2002):

$$\theta(z) = C e^{\frac{-z}{\lambda}} \cos\left(\frac{z}{\lambda}\right), \tag{9}$$

where

$$\lambda = \left(\frac{4 T_0 K_m K_h}{\gamma g \sin^2(\alpha)}\right)^{1/4}. \tag{10}$$

The pre-factor, C, is the temperature perturbation at the glacier surface, which in our case is the difference between surface temperature (273.16 K) and the temperature of the atmosphere at 100 m above the surface. The quantity $\lambda$ is the natural length scale of the flow with $K_m = K_h = 0.1$ m$^2$s$^{-1}$ the eddy diffusivity for momentum and heat, $\gamma$ the vertical temperature lapse





rate, $\alpha$ the terrain slope, and $T_0 = 280$ K the characteristic temperature. The temperature field in the lowest 50 m is further perturbed by random fluctuations of 0.1 K. The wind field, $U$, is uniformly initialized with 8 ms$^{-1}$ throughout the domain.

## 2.4 Boundary Conditions

The lateral boundaries are specified as periodic. At the top boundary a no-slip zero-stress boundary is used. The pressure gradient is set based on the boussinesq density gradient normal to the boundary, and the potential temperature gradient is specified according to the initial profile. At the surface, the same pressure boundary condition is used as at the top boundary.

The filter and grid resolution are too coarse to resolve the near-wall motions, including in the viscous wall region, so that their influence are modelled by some sort of model. A local version of Schumann's shear stress model is applied at the surface (Churchfield et al., 2014; Schumann, 1975; Wan et al., 2007). The Reynolds stress tensor is zero except for the off-diagonal components $\boldsymbol{\tau}_{13}$ and $\boldsymbol{\tau}_{23}$, with

$$\boldsymbol{\tau}_{13} = -u_*^2 \frac{\overline{\boldsymbol{U}_x}(z_1)}{|\overline{\boldsymbol{U}}(z_1)|}, \qquad\qquad \boldsymbol{\tau}_{23} = -u_*^2 \frac{\overline{\boldsymbol{U}_y}(z_1)}{|\overline{\boldsymbol{U}}(z_1)|}, \qquad\qquad (11)$$

where $u_*$ is the friction velocity, $z_1$ the height of the first cell centre adjacent to the wall, and $||$ denotes the magnitude of the local velocity parallel to the surface. To solve for the unknown friction velocity the Monin-Obukhov scaling law is used. The details of the optimization are discussed by Churchfield et al. (2014). Strictly speaking the Monin-Obukhov scaling law neither applies to complex terrain nor is it formulated to apply the laws locally (Stoll and Porté-Agel, 2006; Wan et al., 2007). However, there is as yet no better solution to solve this problem.

The surface temperature flux is determined using Monin-Obukhov scaling laws for velocity and potential temperature (Basu et al., 2008). At each time step the surface temperature is updated according to the heating rate (see Section 2.5).

## 2.5 Numerical Experiments

Given the large computational costs, the analysis is confined to four pseudo-realistic case experiments. The simulations merely differ in the geostrophic flow direction ($0°$, $90°$, $180°$ and $270°$). For the simulations, a constant pressure gradient is imposed to drive the geostrophic wind velocity of 8 ms$^{-1}$ at a height of 5500 m. The aerodynamic roughness height, $z_0$, is set to 0.1 m for the land surfaces and to 0.001 m for the glacier surface, respectively. The glacier surface temperature is kept constant at the melting point during the simulation. The surrounding topography is heated with a constant heating rate of 1.2 K hr$^{-1}$. The atmospheric background state for temperature and pressure is derived from ERA-Interim data (Dee et al., 2011) from the 12th August 2013 (see Section 2.3 for details). The selected day had clear skies apart from some isolated orographic clouds at the ridge south of Fürkeleferner. The LES model is integrated for 8 hours. The mean quantities and statistics are calculated from the last hour.





## 2.6 Numerical Mesh

Besides the fluid dynamical challenges, the numerical model must be able to cope with complex topography. The OpenFOAM solver allows for unstructured grids, which can be adapted more easily to steep topography than commonly used terrain following grids. The 3-dimensional unstructured mesh is generated with the OpenFOAM utility snappyhexmesh. The tool auto-matically generates hexahedra and split-hexahedra meshes from triangulated surface geometries, i.e. Digital Elevation Models. In this study, the mesh is generated from a high-resolution elevation model (1 m horizontal resolution) derived from airborne laser-scans conducted in September 2013 (Galos et al., 2015). The horizontal extent of the computational domain is 10 km x 10 km and is centred over the Zufallferner. The size should be sufficient to resolve the main scales that are involved in the turbulent energy generation. The domain top is set to 10 km. In order to use periodic boundary conditions opposite DEM boundaries are mirrored. This has been done by setting opposite grid points to their mean value, and slowly relaxing (exponen-tially) the adjacent grid points in the inner domain using a spline algorithm. The resulting relaxation zone has a width of 2 km (160 grid points), which is sufficiently smooth to avoid numerical instabilities. Starting with an initial coarse hexahedra mesh, snappyhexmesh refines the cells closed to the DEM surface by cell splitting and iteratively morphing the split-hex mesh to the surface. The inital isotropic background mesh has been set up with a grid spacing of 200 m. In the lowest 300 m the final mesh has a horizontal resolution of 12.5 m. From the meteorological point of view the very SBL with intermittent turbulence faces some challenges. According to the Monin-Obukov similarity theory, the height above the ground $z$ and the Obukhov length $L_s$ are the only relevant scaling variables. The theory is only valid within the surface layer, where the vertical divergence of the fluxes are negligible (variations smaller than 10% of their magnitude). However it turns out, that the vertical gradient of the heat flux above glaciers is normally greatest near the surface. Strictly speaking the Monin-Obukhov theory is not valid when the surface layer is below the observational or model level. This poses the need for a fine mesh closed to the surface to allow adequate resolution of the smaller eddies. In order to better resolve the fluxes and shear stresses directly above the glacier the mesh has been further refined by prismatic inflation layers. The cell centre of the first cell is located 0.6 m above the surface, and the heights of the adjacent cells increase with a constant expansion factor of 1.2. Altogether, the final prismatic layers have a total height of about 30 m.

## 2.7 Averaging and Intermittency

The LES resolves the large energy-containing turbulent structures, so that the output fields are fully turbulent. A given fully turbulent variable, $\widetilde{\phi}$, can be decomposed into the large-scale variation and the subgrid-scale turbulence as

$$\widetilde{\phi} = \overline{\phi} + \phi', \tag{12}$$

where the overbar is the grid cell average. The resolved turbulent contribution $\phi''$ is computed by

$$\phi'' = \overline{\phi} - \langle \overline{\phi} \rangle, \tag{13}$$





where the operator $\langle\ \rangle$ is the averaging time scale. In this study the time scale is chosen to be 1 hour (Mahrt, 2010). The local values of the covariances are calculated as the average of the product of the fluctuations. Applying the Reynolds averaging rules finally lead to

$$\overline{\langle\widetilde{w'}\widetilde{\phi'}\rangle} = \langle\overline{w}\rangle\langle\overline{\phi}\rangle + \langle\overline{w''\phi''}\rangle + \langle\overline{w'\phi'}\rangle, \tag{14}$$

where the terms on the right hand side represent the mean advective, resolved and subgrid-scale turbulent flux. As a general measure of turbulence strength we use the standard deviation of the vertical velocity

$$\sigma_w = (\langle\overline{w'w'}\rangle + \langle\overline{w''w''}\rangle)^{\frac{1}{2}}. \tag{15}$$

Occasional bursting events tend to show a more pronounced tail, so that we use the skewness of the vertical velocity variance as a measure to characterise turbulent mixing events (Mahrt, 2010).

## 3   The pseudo-reality atmosphere

### 3.1   Mean flow patterns and vertical profiles

The following section analyses the mean modelled flow patterns and vertical profiles. The analysis is confined to the atmospheric boundary layer near the glacier surface and the kinematic flow properties affecting it. To better illustrate the characteristics we define four regions of interests (R1-4) as well as four virtual sites on the glacier (Z1-4; see Fig. 2).

Figure 2 shows the mean wind velocity, $\langle\overline{U}\rangle$, at 2 m above the ground for each of the four flow experiments. Apparently the flow accelerates as it passes over the summit ridges (R1), due to the strong pressure gradients between the luv side and the ridge region. After passing the ridge the higher pressure on the lee side slows down the flow again. The mean wind velocity at ridges, which are perpendicular to the synoptic flow, sometimes reaches more than $12\ \mathrm{m\,s^{-1}}$ even though the forcing wind velocity is only $8\ \mathrm{m\,s^{-1}}$. The acceleration partly leads to a flow separation behind sharp ridges (grey dashed lines in Fig. 2), resulting in

a thick trailing wake or bluff body formation. In these regions, strong shears generate turbulence which is an important trigger for vertical mixing events.

On lower wider passes and gaps, the flow follows the topography and modifies the wind systems on the lee side. This is particularly evident at the long stretched glacier divide between Zufallferner and Fuerkeleferner (R1). The large-scale flow enhances the katabatic wind when both wind systems are aligned, but retards it otherwise. Since the glaciers are west-east

orientated, surface wind predominantly accelerates during westerly flow (see R2). More general, katabatic winds in the lee of flat passes or glacier divides are strengthened by the synoptic flow.

In the central part of Zufallferner (R3), the wind velocities considerably vary with the large-scale flow directions. For example, northerly and easterly flow significantly enhance the velocities at the southern boundary of Zufallferner. The local acceleration is the consequence of cross valley circulation triggered by the surrounding topography. The strong positive heat





fluxes at the steep slopes create buoyancy forces that drive the thermal circulations. The associated low pressure at the foot of the slopes entrains air from above. While part of the entrained air merges with the up-slope wind, the other part contributes to the glacier wind. The large-scale flow either suppresses or supports the up-slope wind, and hence the entrainment. The results suggest that the intensity of the cross-valley circulation largely explains the wind variations on Zufallferner.

At the glacier tongue (R4), the large-scale flow hardly affects the surface winds. The katabatic winds gently drain down the glaciers (see Table 1) with velocities ranging between 4.5 and 6.0 m s$^{-1}$. Wind velocities are slightly higher for northerly flow ($\sim$7 ms$^{-1}$). The wind magnitudes are characteristic for mountain glacier during blue sky conditions (e.g. Van den Broeke, 1997; Söderberg and Parmhed, 2006). At Z1 and Z2 the Low Level Jet (LLJ) is consistently found below the lowest 12 m (see Fig. 3). However, the intensity and height of the LLJ vary from case to case. The previously discussed crosswind-circulation and its associated enhanced mass-flux during northerly flow significantly lifts and intensifies the LLJ (see Table 1). Strong valley winds, however, tend to retard the down-slope winds by friction which weakens and lowers the LLJ (e.g. at site Z1, easterly flow). Similar, strong shear associated with a rapid veering of the winds with height can drastically reduce the wind velocity. Such a situation appears within the surroundings of Z4 when the down-slope flows are superimposed by southerly large-scale flow.

The temperature deficit increases towards the glacier tongue and implies a larger forcing to the glacier wind (see Table 1). However the reverse situation is observed, as illustrated in Fig. 3. The intensity and height of the wind maximum decreases down-slope, which somehow contradicts the often observed structure of katabatic flows. The reason for this is the still perceptible influence of the large-scale flow on the katabatic winds down to site Z2. This is evidenced by the fact that no wind maximum is found at the higher sites, Z3 and Z4. Nevertheless, there is a significant positive correlation (0.66) between the height and strength of the LLJ.

## 3.2 Turbulence characteristics and intermittency

Fig. 4 shows the standard deviation of the vertical velocity fluctuations, $\sigma_w$, at 2 m above the ground. Along the ridges turbulence is produced by shears and advected downwind with the flow. Therefore highest values of up to 2.0 m s$^{-1}$ do not jointly appear with high wind velocities, but are rather being found behind the sharp ridges.

At some distance away from the mountain ridges the boundary layer is less turbulent ($\sigma_w < 0.5$ m s$^{-1}$). However, the distributions of $\sigma_w$ at the sites Z1-4 are heavily right-skewed (see Table 1), which is a good indication of occasional mixing events embedded within the turbulence. Several studies observed intermittent turbulent mixing events in the SBL above glaciers and analysed their impact on the surface energy balance (e.g. Cullen et al., 2007; Oerlemans and Grisogono, 2002; Söderberg and Parmhed, 2006; van den Broeke, 1997; Smeets et al., 1998; Munro and Davies, 1978; Hoinkes, 1954; Kuhn, 1978; Munro and Scott, 1989). Single mixing events may have only little impact on the time-averaged quantities, but the intermittent heat supply can be substantial for the melt energy (Oerlemans and Grisogono, 2002; Dadic et al., 2013; Mahrt, 2010; Van den Broeke, 1997). Local turbulent mixing events are driven by the characteristics of local turbulence, submeso motions, and the large-scale flow (Helgason and Pomeroy, 2012; Poulos et al., 2007; Högström et al., 2002). Non-local topographic effects, such





as gap flows or bluff bodies, can favour the probability of periodic occurrence of burst events at a given point on the glacier shapening the local mircometeorological conditions (Söderberg and Parmhed, 2006; Litt et al., 2015).

The vertical mixing of momentum and heat is a non-stationary process with changing frequency and intensity across the time (Torrence and Compo, 1998; Roesch and Schmidbauer, 2014). Therefore, it is convenient to analyse the frequency structure

of recurrent intermittency by decomposing the temperature signal into time-frequency space using wavelets. To illustrate the characteristics of intermittency we have calculated the wavelet power spectrum of the temperature signal at Z2 (southerly flow). Fig. 5 shows the normalized wavelet spectrum and the average power taken over time. The global spectrum shows that most of the power is concentrated around 90 s. There are variations in the frequency of occurrence and amplitude of the mixing events. On average episodic mixing events occur every 10 minutes and last for about 90 s. The wavelet spectrum differs at each site

and flow (not shown), but characteristic events are present in all cases. The frequency structure of the recurrent mixing events implies that the surface layer is episodically affected by anisotropic large-scale eddies. Contrary to near-neutral conditions, integral turbulence scales differ significantly between the horizontal and vertical components (see Tab. 1). The scales are in the same order of magnitude as those found by other studies (e.g. Litt et al., 2015; Söderberg and Parmhed, 2006). Since the stable stratification is weak in the surface layer ($Ri_b \sim 0.04$), it is very unlikely that the downward directed heat flux is strong enough

to explain these large differences. More important, probably, is the distortion of the detached eddies (turbulence) by local shear in the surface layer, which leads to groups of elongated sloping eddies (Högström et al., 2002).

### 3.3 Spatial variations of the surface sensible heat flux

According to the principle of energy conservation the local change in the potential temperature tendency of dry air at any given point is related to the advective, turbulent and the radiative heat fluxes. The latter one is not explicitly modelled in this study but

indirectly given by the prescribed surface temperature. In this case, the heating and cooling of the near-surface layer is only a result of the advective and turbulent transport. Local advection is usually negligible over flat terrain and weak wind conditions, but is considered a relevant process on mountain glaciers with consequences on the spatial variations of the surface heat flux (Moore and Owens, 1984).

Fig. 6 shows the spatial variability of the modelled mean surface sensible heat flux over the glaciers. The fluxes vary locally

between 10 W m$^{-2}$ and 120 W m$^{-2}$, with slightly smaller values in the higher parts of the glacier due to lower temperatures. Note that positive signs indicate downward directed fluxes. Along the glacier centre lines the heat fluxes are in the range of 20 W m$^{-2}$ to 60 W m$^{-2}$, which is in good accordance with observations made on mid-latitude glaciers during clear sky conditions (e.g. Giessen et al., 2008; Oerlemans and Klok, 2002; Greuell and Smeets, 2001; Brock et al., 2000). Enhanced heat fluxes occur in the peripheral zones of glaciers and along narrow and deeply carved valleys (e.g. Langenferner, R2), where

strong cross-valley circulations locally advect air towards the glacier (see Fig. 7). The glacier topography locally inhibits a far-reaching advection and restrict the zone of influence to a narrow band along the glacier margin (e.g. R3). Accordingly, the peripheral glacier zones show the highest variability.

Between the individual experiments the spatial variability show striking differences (see Fig. 6). However, these differences are small or even negligible when taking the glacier-wide averages (see Tab. 2). These findings have important implications





on glacier mass balance studies. On the one hand side, distributed mass balance estimates (models) require a fundamental understanding of the heat advection (Fig. 7), since the heat flux can make over 50% of the total energy during large melt events on mountain glaciers in summer (e.g. Cullen and Conway, 2015; Gillett and Cullen, 2011; Van den Broeke, 1997; Hock, 2005; Klok and Oerlemans, 2002; Oerlemans and Klok, 2002; Giessen et al., 2008; Moore and Owens, 1984). On the other hand,

however, we can conclude that topographic effects are less crucial for mean glacier-wide mass change estimates although off-course the calculated amount of total ablation can depend on the spatial (altitudinal) distribution of the sensible heat flux since additional energy causes more melt in areas where the surface temperature is at the melting point. While in areas with surface temperatures lower than the melting point more energy is consumed by the ground heat flux to warm up the glacier.

While advection is essential for local estimates the question remains whether the impact of recurrent mixing events are of

10 the same order of magnitude (see Sec. 3.2). Although the intermittent events temporarily increase the surface heat flux, there is little evidence that these events impact the time averaged fluxes. In conclusion, local thermal micrometeorological conditions are mainly shaped by warm air advection through the cross-valley circulations.

### 3.4 Reliability of the LES experiments

Even though the pseudo-reality atmosphere seems to describe realistically the physical processes and patterns, the simulations

must be interpreted with care. The patterns depend on the model assumptions which include parametrizations and idealized boundary conditions.

A crucial assumption is the surface roughness length. To obtain more general results, uniform values of $z_0$ for snow and ice with 0.001 m are used, which is in the range of commonly used values (e.g. Braithwaite, 1995; Giessen et al., 2008; Brock et al., 2000; Hock, 2005; Greuell and Smeets, 2001). The "uniform" assumption ignores temporal and spatial roughness length

variations. However, potentially such variations can have a strong influence on the magnitude of the surface energy fluxes (Brock et al., 2000; Giessen et al., 2008). We argue that this assumption is acceptable for the summer season, and in particular for the end of the ablation season. In this time of the year, the spatial variability of $z_0$ is usually small and almost similar values can be found for snow and ice (Brock et al., 2000).

The roughness lengths of snow and ice are relatively small compared to non-uniform roughness elements at a scale of tens

of meters such as deep seracs or ice falls. The scales of these elements are approximately of the same order as the horizontal model resolution. Enhanced mixing due to the sudden roughness changes is therefore not resolved by the model, and it is very likely that the model underestimates the overall variability.

In general, the model resolution is very decisive for the overall quality of the LES simulations. LES require that $\sim 80\%$ of the turbulent kinetic energy is resolved by the model itself, and only a minor part is modelled by the SGS model (Pope, 2000).

In the performed experiments, on average 20-30% of the total turbulent kinetic energy is modelled by the SGS model. Slightly higher fractions, of up to 40%, are found at exposed mountain ridges. The used Lagrangian-averaged dynamic Smagorinsky model assumes that the energy transfer from the resolved-scale eddies to the residual motions is entirely balanced by the dissipation of kinetic energy. However, dissipation is not necessarily in balance with the energy production in stably stratified boundary layers. As a consequence, the SGS model is likely to dissipate too much energy. In case of the four experiments the





stable stratification was weak ($Ri_b \sim 0.04$), and we can assume that the overestimated dissipation is negligible. Which SGS model works best for stable boundary layers is not easy to tell, but the Lagrangian-averaged SGS model seems to work well in our study.

## 4   Estimation of the energy exchange using the Bulk-Approach

Physically based distributed mass balance models are often applied to translate the local-scale weather conditions into net mass gain and loss at the glacier surface. The ablation process, which removes ice and snow, is controlled by the net energy balance at the ice-atmosphere interface. Direct measurements of energy balance components exist in most cases only for radiation, while surface heat and moisture fluxes are rarely measured directly on glaciers. The simplest and most widely used method to parametrize the turbulent energy exchange from available meteorological observations is the bulk approach. The approach is based on the Monin-Obukhov theory and assumes constant fluxes within the surface layer. This is not necessarily true in the presence of a LLJ, but the method is found to give good results when measurements are below the wind velocity maximum (Greuell and Smeets, 2001). The surface sensible flux is usually estimated by

$$Q_H = \frac{\rho c_p \kappa^2 U (T_a - T_s)}{\left[\ln\left(\frac{z}{z_0}\right) + \psi_m\left(\frac{z}{L_s}\right)\right]\left[\ln\left(\frac{z}{z_{0h}}\right) + \psi_h\left(\frac{z}{L_s}\right)\right]}, \tag{16}$$

where $\rho$ is the air density ($\mathrm{kg\,m^{-3}}$), $c_p$ the specific heat of air at constant pressure ($1004\,\mathrm{J\,kg^{-1}\,K^{-1}}$), $\kappa$ is the von Karman constant (0.4), $U$ is the wind velocity ($\mathrm{m\,s^{-1}}$), $T_a$ and $T_s$ the air temperature (K) at the height $z$ (m) and the surface. The parameters $z_0$ and $z_{0h}$ (m) are the roughness lengths for momentum and heat, respectively. The characteristic length scale $L_s$ (m) is the Obukhov-length and is proportional to the height of the dynamic sub-layer. The vertically integrated stability functions for momentum, $\psi_m$, and heat, $\psi_h$, are given as

$$\psi_m = \psi_h = \frac{4.7 \cdot z}{L_s}. \tag{17}$$

It is straightforward to apply Eqn. 16 to any given point on the glacier, given that all quantities are known. However, highly resolved observational data on glaciers, as required to characterise the spatial fields, are usually scarce and need to be extrapolated. Extrapolation algorithms in turn are based on simplified distribution assumptions and are unlikely to sufficiently reconstruct the full variability of a quantity in time and space. To identify the shortcomings in the local sensible heat flux estimates due to deficiencies in the observations (extrapolation), we consider the LES as pseudo-reality. The pseudo-reality atmosphere is not required to be an observed real world case, but needs to be plausible in the sense that relevant processes are realistically simulated. As demonstrated in Sec. 3.1, 3.3 and 3.2, the LES model captures the relevant processes observed in mountainous terrain.





### 4.1 Shortcomings in the sensible heat flux estimates related to the large-scale flow direction

To illustrate how the flux estimates depend on the local flow conditions, we defined two virtual observation points at Zu-fallferner ($Z_0$ and $Z_a$; see Fig. 8). The pseudo-observed wind velocities and temperatures at the two sites were linearly extrapolated across the glacier (e.g. Paul and Kotlarski, 2010; Machguth et al., 2009; Huintjes et al., 2015; Weidemann et al., 2013;

Jarosch et al., 2012). Assuming that the Obukhov-length, $L_s$, equals the observed value at $Z_0$, the surface heat flux was then calculated at all grid points (Eqn. 16).

Fig. 8 shows the differences between the calculated surface heat fluxes obtained from the bulk method and the LES. As $T_s$, $z_0$ and $L_s$ are known from the LES, the discrepancies must be the result of the insufficient characterisation of the spatial $U$ and $T_a$ fields. It is evident that the forcing fields lack to reflect the variability of the local processes which originate from the

complex topography. Shortcomings are eminently striking in regions of warm air advection (see Sec. 3.3 and Fig. 7). The bulk approach, for instance, underestimates the fluxes by up to 40 W m$^{-2}$ in the peripheral zone of Zufallferner (steep slopes) and also in the vicinity of $Z_a$. Local advection processes equally explain the deficits in the higher regions of Fürkele Ferner.

On contrary, the fluxes are largely overestimated along the glacier centrelines and tongues. In these regions the well-developed katabatic flow prevents warm air advection from the surroundings (see Sec. 3.1). For example, this is the case

at the tongues of Fürkele Ferner and Zufallferner where glacier winds converge due to the topography. Here, the persistent winds are barely perturbed by the warm air advection from the surrounding terrain. Instead, the air continuously cools on the way down the glacier by a downward sensible heat flux, and is therefore potentially cooler than in other parts of the glacier. The temperature gradients, determined from the two locations ($Z_0$ and $Z_a$), do not account for this additional cooling. Hence, the extrapolated fields are too warm and the bulk approach overestimates the surface heat fluxes by 10-30 W m$^{-2}$.

On a glacier-scale, the bulk approach underestimates the average heat flux by up to 7 W m$^{-2}$ (see Tab. 2). The only exception is for the southerly case where local differences almost cancel each other out.

### 4.2 Shortcomings in the sensible heat flux estimates due to the choice of observation sites

The choice and number of observation sites on glaciers is always a compromise between logistic feasibility, financial expenditure and scientific issue. These factors usually restrict the monitoring activities to a few sites along the glacier centrelines.

Even from a pure scientific perspective the choice of observation sites that meet all requirements is challenging.

To explore how the choice of observation sites influences the spatial variation of the heat flux estimates, we define a set of virtual observation on Zufallferner ($Z_0$ and $Z_{a-d}$; see Fig. 9 and Tab. 3). For each combination of $Z_0$ and $Z_{a-d}$ the heat fluxes are estimated according to Eq. 16.

Fig. 9 shows the differences between the bulk estimates and pseudo-reality atmosphere. Evidently, the bulk estimates lack to

reflect the variability in time and space. Since the spatial patterns are similar for all cases, the shortcoming must be related to the insufficient characterisation of the temperature advection process. The magnitude of the differences, however, result from the derived temperature gradients, and thus on the location of the second station, $Z_{a-d}$. In the case that stations are located in a region of strong temperature advection (e.g. case $Z_0$-$Z_a$ and $Z_0$-$Z_b$) the derived temperature gradient is too large, and the





bulk approach overestimates the surface heat fluxes in most regions of the glacier. Similar, temperature gradients are too small when stations are protected from warm air transport, and on average fluxes are underestimated (e.g case $Z_0$-$Z_c$ and $Z_0$-$Z_d$). On glacier-wide average the excess/deficit in energy varies between -14.5 and 16.6 $\mathrm{Wm^{-2}}$ (see Tab. 3).

The results confirm that the phenomenological understanding at few locations and weather situation is not valid beyond the
case and insufficient to infer on the micrometeorological conditions on mountain glaciers. We assumed a uniform wind velocity which ignores important local flow features (e.g. gaps and bluff bodies) and drastically underestimates the wind velocities over large areas. Like in most other mass-balance studies, the temperature fields were decoupled from the flow fields what generates a static temperature field without allowing for temperature advection.

## 5  Conclusions

We have shown how complex topography influences the micrometeorlogical conditions on three mid-latitude mountain glaciers in the Italian Ortler-Cevedale Group. The idealized LES experiments demonstrate that heat advection associated with the wind systems shape the thermal conditions on the glaciers in clear sky conditions during summer. In particular, the cross-valley circulations, and bluff body formations behind sharp ridges, transport warm air from the surroundings to the peripheral zones of the glaciers and locally increase the surface sensible heat fluxes by 50-100 $\mathrm{Wm^{-2}}$. Intermittent downburst events, however,
entrain little heat from the free atmosphere towards the surface. The effective energy surplus is supposed to be even higher when the longwave radiation is parametrised by air temperature.

Our pseudo-reality experiments demonstrate that it is challenging to fully characterise the micrometeorological conditions over glacier surfaces from a limited number of observations. Linearly extrapolated forcing fields fail to reflect the temperature variability that originates from insufficient characterisation of advection. The shortcomings in the forcing fields have direct
consequences on estimated surface heat fluxes (e.g. by the bulk approach). Local errors in the surface heat fluxes of up to 60 $\mathrm{Wm^{-2}}$ are strong enough to significantly affect the ablation rate estimates as well as the derived climate sensitivities of mountain glaciers.

The choice of observations sites, and thus the derived temperature gradients, determine the magnitude of the local heat flux errors. Calculated temperature lapse rates are steeper ($< -0.01$ $\mathrm{Km^{-1}}$) than the environmental lapse rate ($-0.0065$ $\mathrm{Km^{-1}}$)
when one of the stations is influenced by warm air advection. Consequently, the overestimated air temperatures produce higher downward directed heat fluxes for most parts of the glaciers. In case stations are protected from warm air transport or located in well developed katabatic flows, calculated temperature gradients are generally shallower ($> -0.005$ $\mathrm{Km^{-1}}$) than the environmental lapse rate. The shallow lapse rates result in very low heat flux estimates in the peripheral and higher zones of the glaciers, where heat advection is an important process. As a glacier-wide average, the choice of observation sites causes
errors of about $\pm 16$ $\mathrm{Wm^{-2}}$ ($\sim 20\%$). The estimated errors are considered conservative given the weak geostrophic forcing and low surface heating rate. However, the error quantification is only valid for the specific experimental design, and the infinite topographic possibilities and variety of site combinations make it impossible to draw a general conclusion about the best sites on a glacier.



We can conclude that a profound knowledge of the heat advection process is needed when small-scale variations of surface energy balance are required. Current thermodynamic and statistical centreline models describe temperature variations along the flowline of glaciers, but do not resolve the cross-glacier variability (e.g. Greuell and Böhm, 1998; Carturan et al., 2015; Ayala et al., 2015; Petersen et al., 2013). In order to account for the lateral variations, temperature and wind fields need to

5  be coupled. We suggest that future efforts should consider more representative wind fields (e.g. simulated by mass-consistent models) in concert with simple centreline models using off-glacier stations.

*Acknowledgements.* We gratefully acknowledge financial support by the Deutsche Forschungsgemeinschaft (DFG), no. SA 2339/4-1. This work was also supported and partly financed by the Autonome Provinz Bozen - Südtirol, Abteilung Bildungsförderung, Universität und Forschung.



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



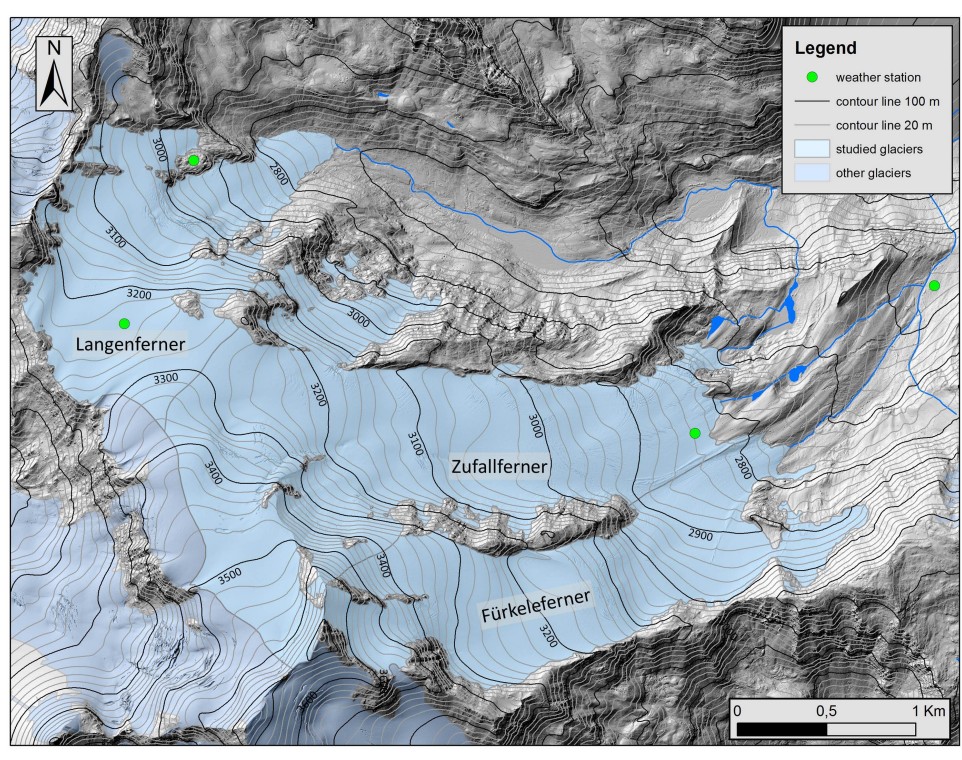

**Figure 1.** Map showing the surface topography of the studied glaciers and the surrounding terrain.





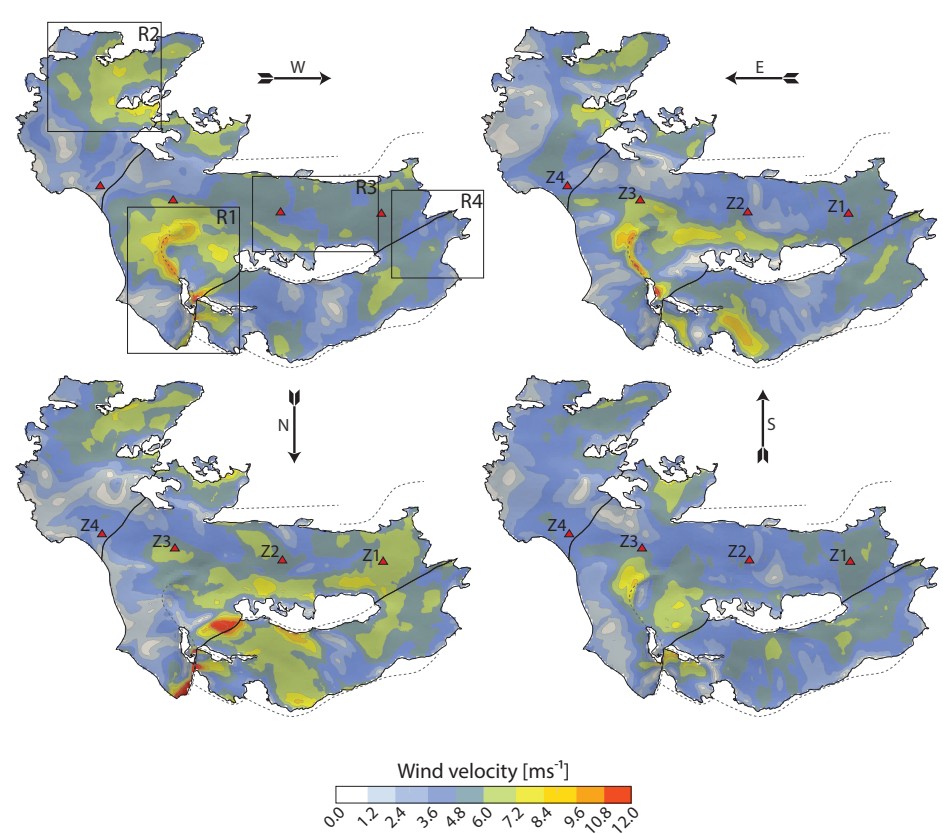

**Figure 2.** Mean velocity of the surface wind fields (2 m) for each of the four case experiment. The four boxes R1-4 and the sites Z1-4 define regions and locations on the glacier which are used for discussion in the results section. The grey dashed lines represent sharp ridges in the study area.





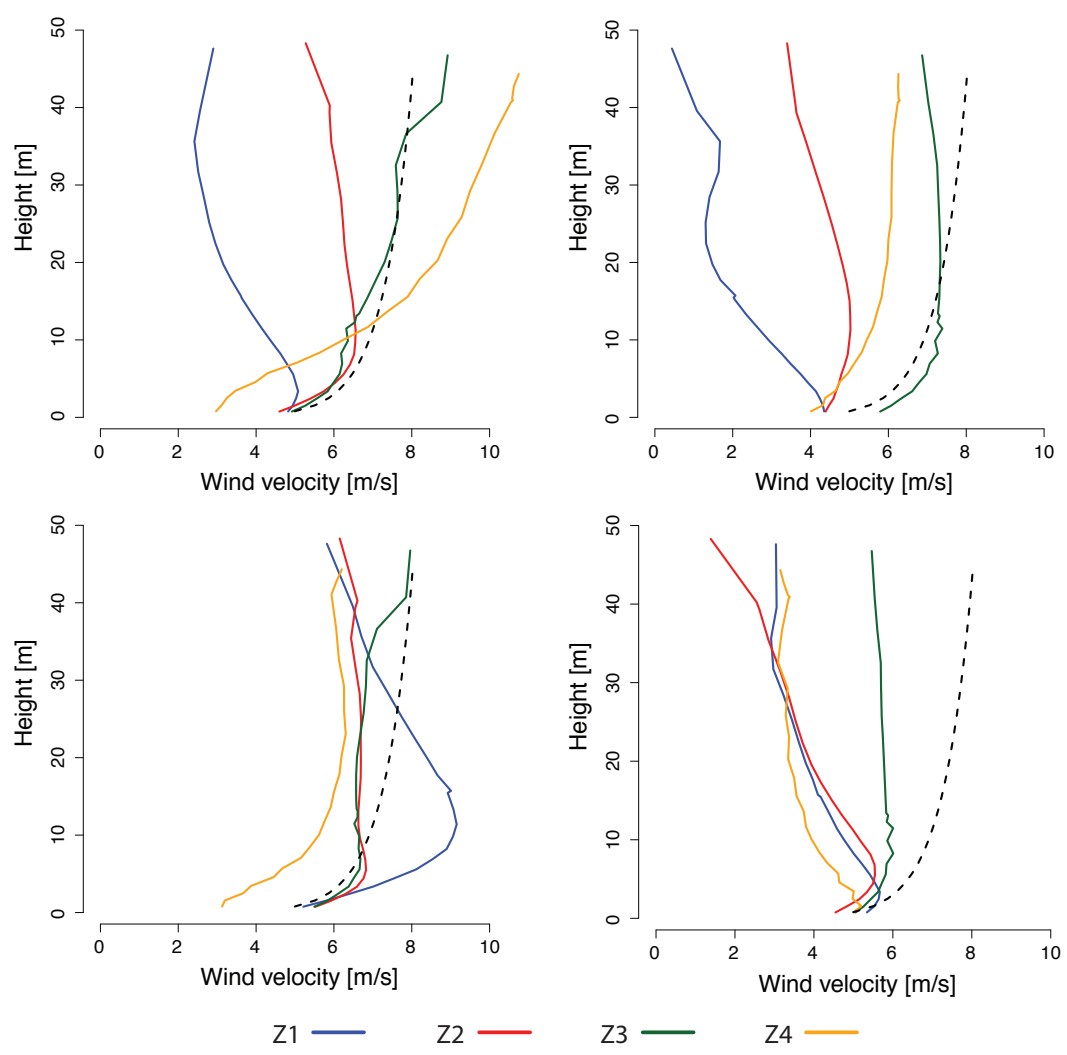

**Figure 3.** Vertical profiles of the mean wind velocity at the four sites (Z1, Z2, Z3 and Z4) for each case experiment. The dashed line represents a neutral logarithmic wind profile with $z_0 = 0.01$ m and $u_* = 0.3$ ms$^{-1}$.





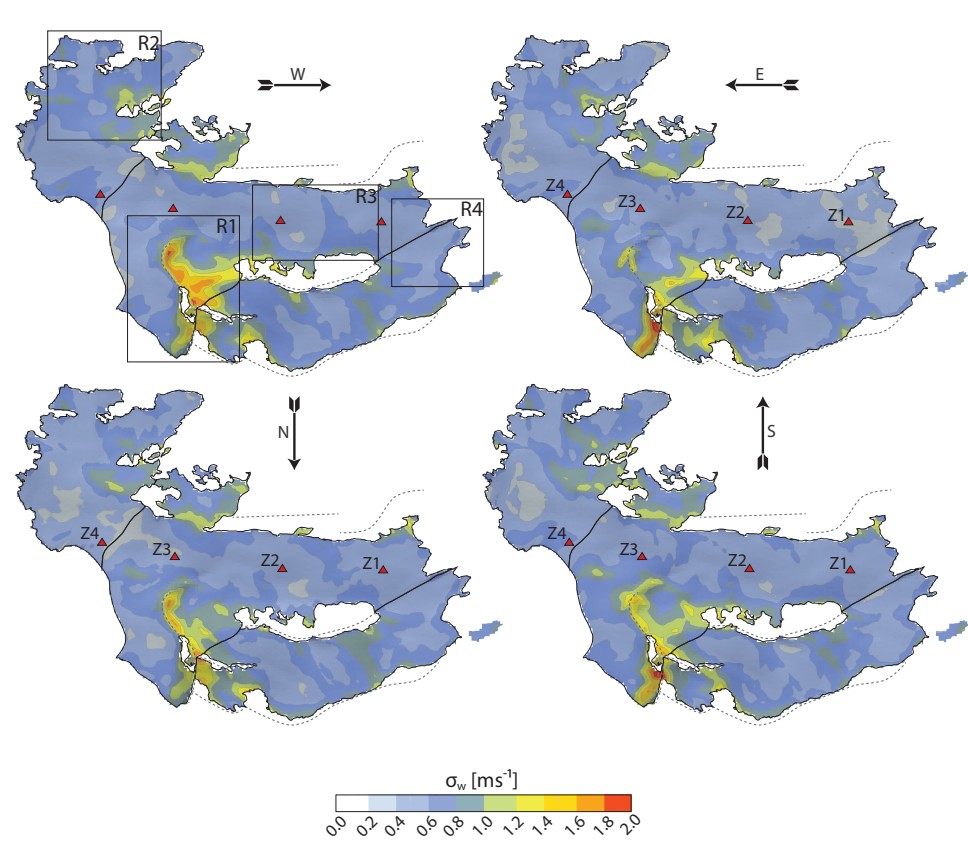

**Figure 4.** Standard deviation of the vertical velocity fluctuation at 2 m above ground for each of the four case experiments.





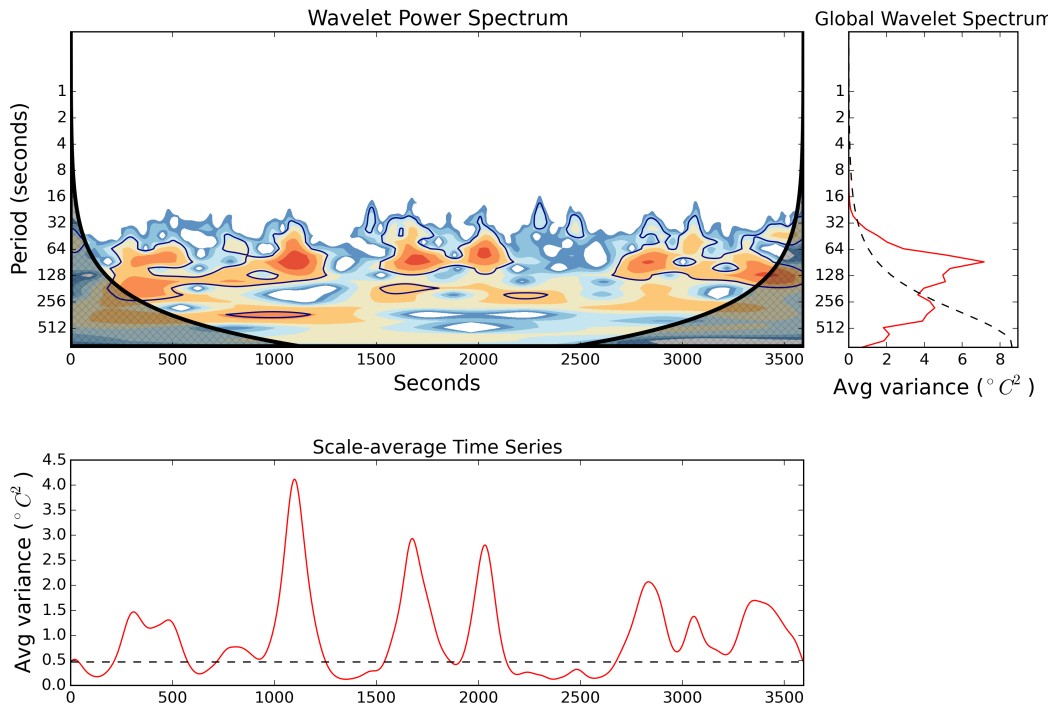

**Figure 5.** Example of a rectified wavelet power spectrum of the temperature signal at location Z2 for southerly flow (upper left column), the time average-wavelet power spectra (right column), and the scaled-averaged time series (lower left column). Red and blue indicate high and low scaled powers (in base 2 logarithm), respectively. Black lines outline the wavelet spectrum at a 95% confidence level. The cross-hatched region marks the cone of influence, where edge effects become important.



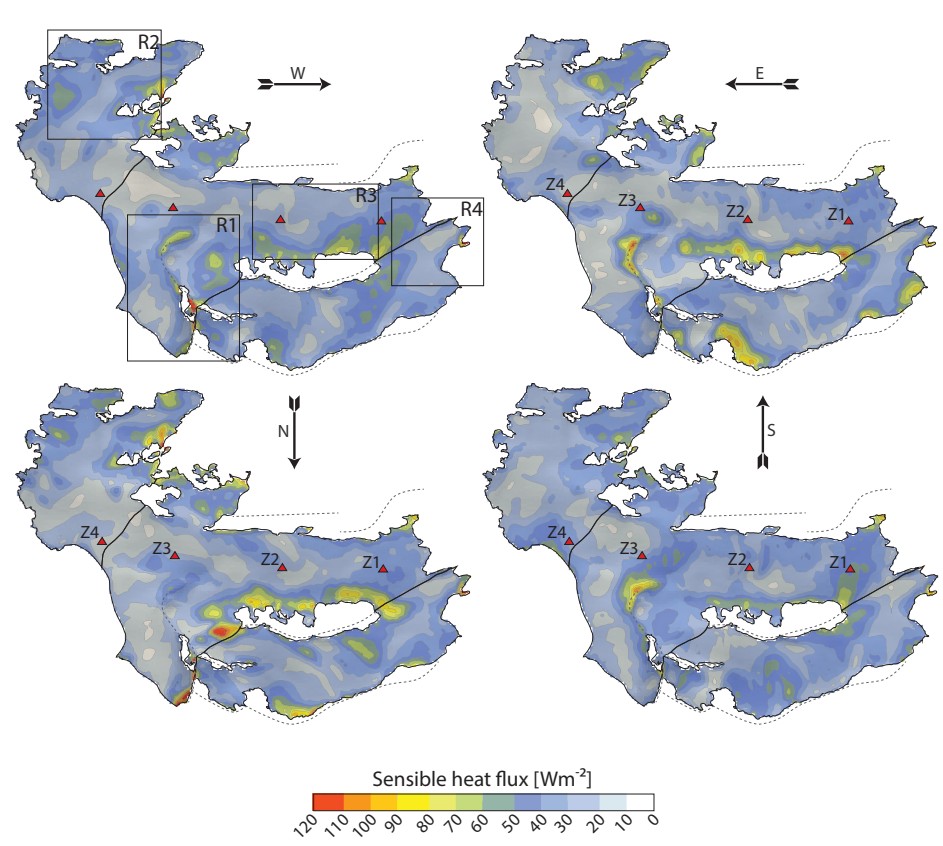

**Figure 6.** Mean sensible heat flux from the LES runs for each of the four case experiments.




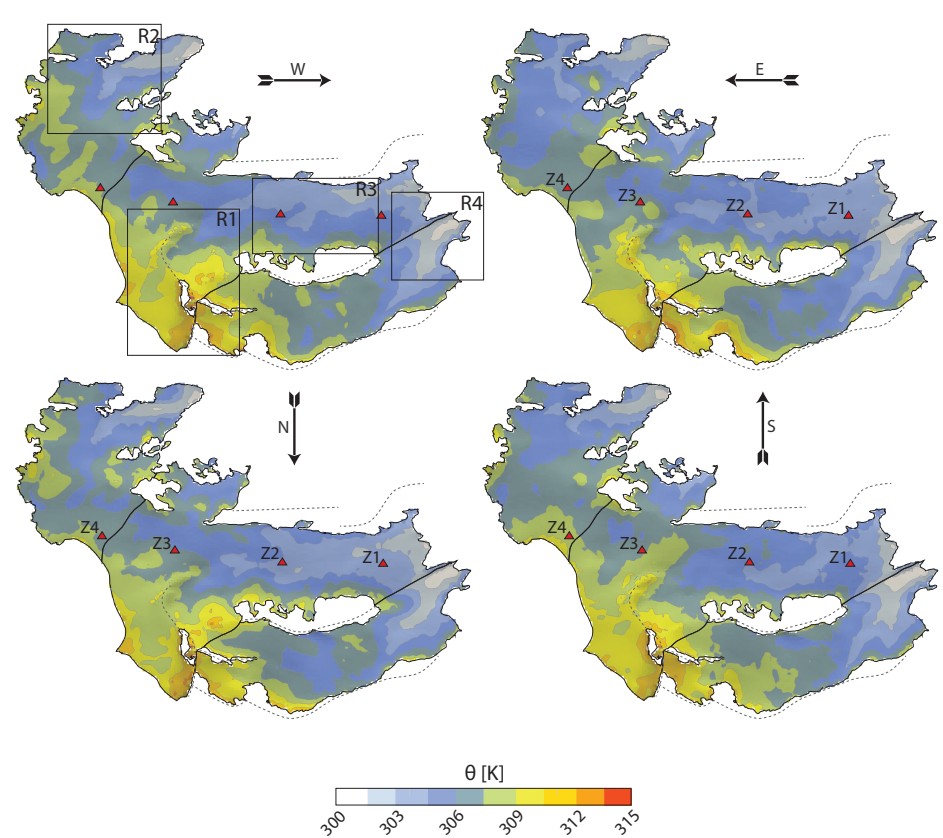

**Figure 7.** Potential temperature at 2 m above the surface for each of the four case experiment. The four boxes R1-4 and the sites Z1-4 define regions and locations on the glacier which are used for discussion in the results section. The grey dashed lines represent sharp ridges in the study area.





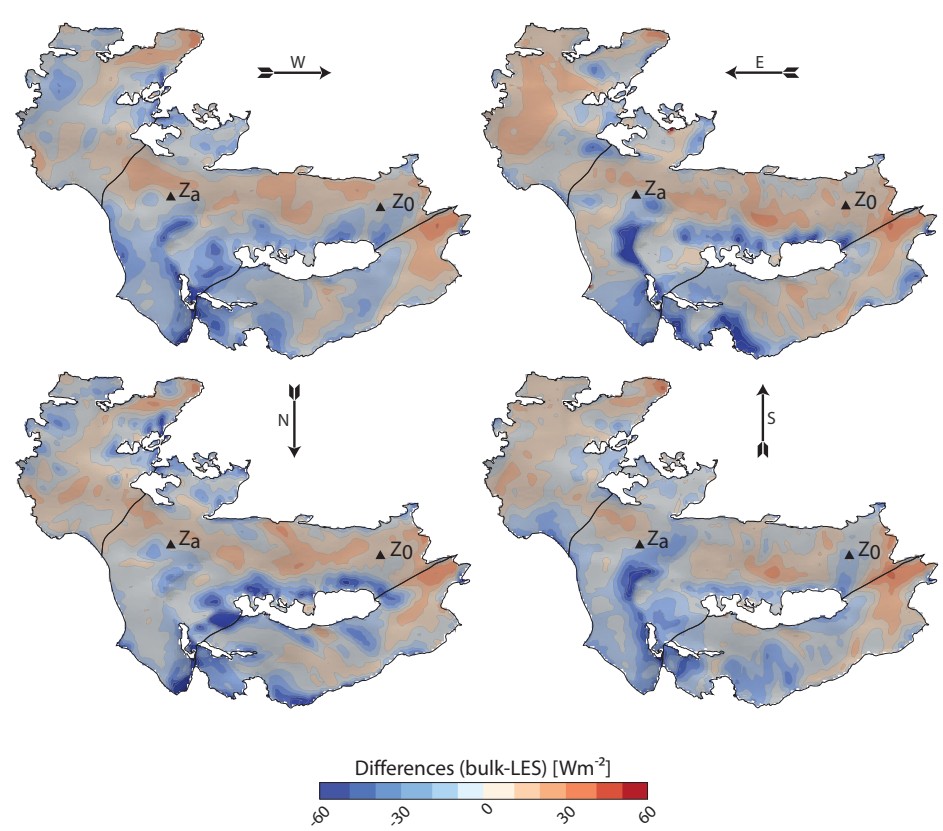

**Figure 8.** Differences in the mean surface sensible heat fluxes between the LES and the bulk method for different wind direction. Positive differences correspond to an overestimation of the surface heat flux by the bulk approach.




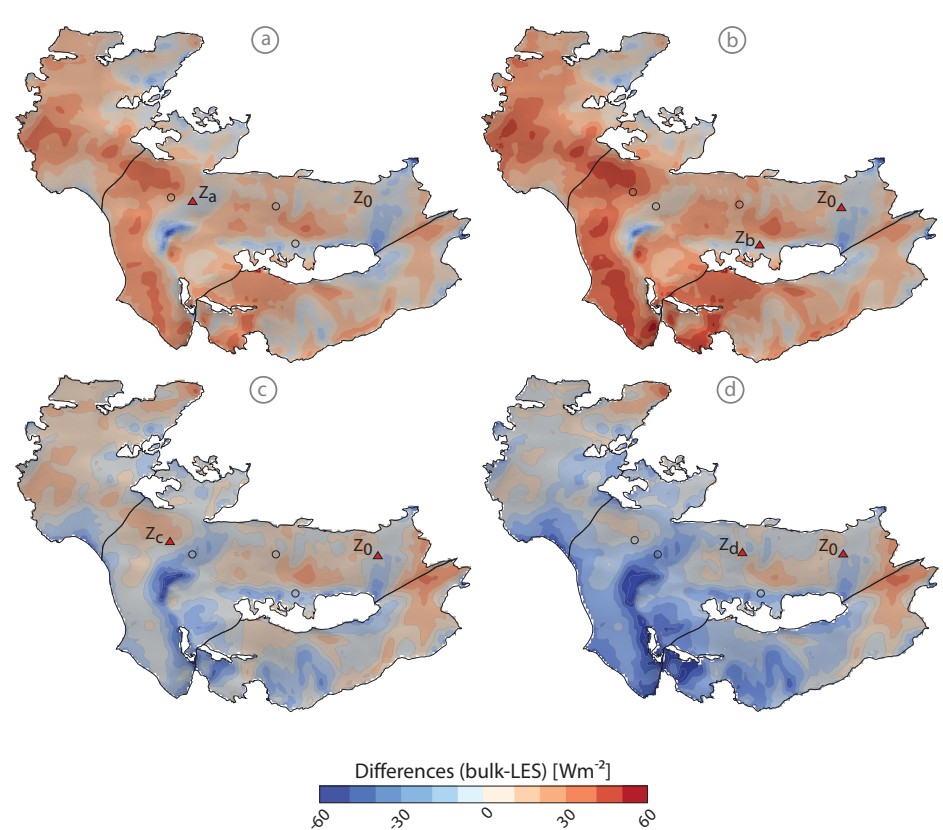

**Figure 9.** Shown are the differences in the surface sensible heat fluxes between the LES and the bulk method. The cases A-D indicate different extrapolation scenarios based on the pseudo-observations ($Z_0$, $Z_{a-d}$) . Positive differences correspond to regions where the fluxes have been overestimated by the bulk approach.





**Table 1.** Mean statistics at the sites (Z1-4) at 2 m above the surface, derived from the Large-eddy simulations. Given are the heights of the LLJ (second column), wind velocities of the LLJ (third column), wind directions (fourth column), differences between the surface temperature and the temperatures at 100 m above the ground (fifth column), skewness of the vertical velocity variances (sixth column), integral turbulence scales (seventh to ninth column), and the bulk Richardson numbers (last column).

| Experiment/Location | h [m] | v [ms$^{-1}$] | dir [°] | $\theta_\Delta$ [K] | Skewness | $\frac{\sigma_u}{u_*}$ | $\frac{\sigma_v}{u_*}$ | $\frac{\sigma_w}{u_*}$ | $Ri_b$ |
|---|---|---|---|---|---|---|---|---|---|
| West / Z1 | 3.3 | 5.0 | 235 | 13.3 | 2.35 | 4.96 | 4.54 | 0.96 | 0.04 |
| West / Z2 | 11.2 | 6.5 | 252 | 12.6 | 6.94 | 3.96 | 4.57 | 0.84 | 0.03 |
| West / Z3 | - | - | 203 | 10.8 | 3.14 | 3.41 | 2.70 | 0.54 | 0.02 |
| West / Z4 | - | - | 187 | 10.5 | 4.73 | 4.99 | 3.35 | 1.24 | 0.04 |
| East / Z1 | 0.8 | 4.3 | 254 | 11.6 | 3.12 | 2.79 | 4.11 | 0.53 | 0.04 |
| East / Z2 | 11.2 | 5.0 | 266 | 11.0 | 2.26 | 3.40 | 6.05 | 0.64 | 0.03 |
| East / Z3 | 11.4 | 7.3 | 164 | 9.6 | 4.70 | 3.51 | 2.97 | 0.63 | 0.03 |
| East / Z4 | - | - | 234 | 9.2 | 3.33 | 3.89 | 4.67 | 1.26 | 0.03 |
| North / Z1 | 11.3 | 9.1 | 240 | 13.1 | 3.11 | 4.26 | 6.99 | 0.91 | 0.03 |
| North / Z2 | 5.5 | 6.8 | 275 | 10.9 | 6.03 | 3.74 | 4.83 | 0.66 | 0.03 |
| North / Z3 | - | - | 225 | 9.8 | 6.05 | 3.14 | 2.43 | 0.47 | 0.02 |
| North / Z4 | - | - | 234 | 9.5 | 2.72 | 4.91 | 3.42 | 1.05 | 0.05 |
| South / Z1 | 3.3 | 5.6 | 255 | 12.7 | 2.60 | 3.55 | 4.46 | 0.83 | 0.04 |
| South / Z2 | 5.5 | 5.5 | 290 | 11.2 | 2.90 | 4.13 | 4.41 | 0.77 | 0.03 |
| South / Z3 | 8.2 | 6.0 | 159 | 10.8 | 2.29 | 6.16 | 3.93 | 1.26 | 0.04 |
| South / Z4 | 1.5 | 5.1 | 163 | 10.0 | 2.09 | 4.63 | 5.27 | 1.32 | 0.05 |

**Table 2.** Comparison of the bulk approach with the LES for distinct flow directions. Positive mean relative errors correspond to an overestimation of the fluxes by the bulk approach.

| | West | East | North | South |
|---|---|---|---|---|
| Large-eddy simulation [Wm$^{-2}$] | 33.8 | 31.2 | 33.8 | 33.0 |
| Bulk approach [Wm$^{-2}$] | 26.9 | 26.0 | 27.4 | 33.8 |
| Mean relative error [%] | -20.3 | -16.6 | -19.3 | 2.2 |





**Table 3.** Shown are the mean glacier-wide sensible heat fluxes using the bulk approach with linearly extrapolated temperature and wind fields. The table shows extrapolation scenarios based on different pseudo-observations ($Z_0$, $Z_{a-d}$). The exact location of the pseudo-observations is given in Fig. 9. Given are also the mean differences between the bulk estimates and LES. Positive differences correspond to an overestimation of the fluxes by the bulk approach.

|  | $Z_0$-$Z_a$ | $Z_0$-$Z_b$ | $Z_0$-$Z_c$ | $Z_0$-$Z_d$ |
|---|---|---|---|---|
| Bulk approach [$\mathrm{Wm}^{-2}$] | 42.76 | 49.66 | 28.85 | 18.77 |
| Mean difference [$\mathrm{Wm}^{-2}$] | 9.82 | 16.58 | -4.04 | -14.53 |
| Lapse rate [$\mathrm{Km}^{-1}$] | 0.015 | 0.019 | 0.005 | 0.004 |