# Peer review of "Effects of local advection on the spatial sensible heat flux variation on a mountain glacier"

_The Cryosphere, 2016_

## Referee Comment (RC1) · Anonymous Referee #1 · 12 Jul 2016

General comments

The authors Sauter and Galos present a pseudo-reality case study of meteorological conditions over glaciers in the Bozen Province of Northern Italy. Their work demonstrates, using detailed Large Eddy Simulations (LES), that surplus energy associated with local wind systems produces significant increases in sensible heat fluxes above the glaciers. The study presents comparative results from LES simulations and the commonly adopted bulk approach based on Monin-Obukhov theory as well as highlighting a key problem of site 'representativeness' for forcing data in distributed models. Overall, the work is well written and presented. The approach is novel and emphasises a key problem in the literature which is so often dealt with using over-simplified extrapolation techniques. Recent works have indeed demonstrated how inadequate the extrapolations of temperature from a few sites can be and which often do not account for boundary layer effects or local wind conditions. Although I suggest some minor revisions to the discussions manuscript, I would recommend this work for publication in The Cryosphere due to its scientific quality and significance.

Due to the computationally intensive nature of the LES, it is understandable that a small timeframe is most suitable to demonstrate the expected variation of sensible heat fluxes over the glaciers. However, I think the paper would benefit from having more detail on the conditions of the hour for which statistics are presented. The authors describe a blue sky condition which is known to be favourable for the development of a katabatic boundary layer, however the strength of the boundary layer can also be affected by the ambient air temperature (data from the off-glacier sites seen in Figure 1 could aid this). Furthermore, could the LES be compared with a cooler/cloudier hour? Though adding some extra work, I think this would benefit the scientific community and be informative for when (under which conditions) sensible heat fluxes are most likely to be inadequately modelled.

As the work details, the LES is not required to be an observed real-world case, as the realistic simulation of processes and their spatial variation is key. However the authors indicate several weather stations in Figure 1 (which are not used). It would be interesting to present what the actual lapse rate on glacier would be and also compare the calculation of sensible heat fluxes using this measured data. If no AWS measurements are to be utilised in this study, please remove them from the figure.

The authors also outline several sub-regions and 'virtual' sites of interest on Zufallferner though with no clear justification for why. I think it is important to demonstrate the spatial variation of wind fields along a glacier centreline and focus on specific sites (i.e. Z1-Z4), particularly when attempting to simulate and understand interactions of the glacier boundary layer with synoptic scale winds. Furthermore, the selection of temperature extrapolation locations is important although often somewhat arbitrary in many studies.

[Figure]

However, the presentation of several different sites between figures (Figures 7,8,9 for example) and their naming conventions (Z3 changes to Za then to Zc) is misleading. The authors should add some additional reasoning to their choices of virtual sites. The authors should also guide the reader to aspects of figure subplots by labelling them (i.e. a-d). Misleading information for Figure 3 is particularly noteworthy.

Finally, while it is clear from section 1 what the problems of the literature are (and it is very well written), I think it is important to stress in a little more detail what the aim of the paper is and add some more discussion regarding the applicability of an LES approach at the end.

Specific comments

1 7: Add the temporal scale for which of the flux over- and under-estimates are found (i.e. 1 hour of statistics).

1 18: Re-word "loss of information".

2 10: I think it is important to stress that this "over 50%" contribution from turbulent heat fluxes is typical for overcast conditions or for maritime glaciers (as is given by the studies you cite –e.g. Cullen and Conway, 2015) as otherwise the dominance is typically from shortwave radiation. For your study you assess a clear sky condition and a continental glacier.

2 13: Replace "peculiar" with "particular".

2 25-26: Though I agree that there is still much to be understood about the impact of these assumptions on glacier melt rates, citing some of the work which has made attempts to use distributed temperature for this purpose would be suitable here. For example, Immerzeel et al. (2014) investigate this for a catchment/valley scale and Shaw et al. (2016) investigate this for a debris-covered glacier.

3 19: I assume here that you refer to the surface boundary layer for "SBL"? Write out in full before using the acronym.

[Figure]

3 20: What does SGS refer to? Write out in full as well.

4 4: A minor point, but you are missing an equation number for eddy viscosity (this should be eqn 7).

4 9-11: This sentence needs re-writing. It is unclear what it is trying to say and the sentence has syntax errors.

5 2: Changes in temperature and phase from radiative forcing would be relevant if the LES approach was adopted over a longer time-frame. This may be worth adding to the discussion?

5 16-17: How is the topography representative of many in the European Alps? Can you also add the mean slope of the glacier to this section?

5 21: What grid size do you use for the ERA-Interim reanalysis data? Is this re-sampled from the 6 hourly temporal scale of ERA-Interim? Additional detail would be useful here.

5 28: Specify if the 100 m temperature is that from the ERA-Interim.

6 2: Why 8 m/s-1? Is this the mean value from the given six hour period of the reanalysis data?

6 8: It is unclear what you mean by this - "some sort of model". Please re-word this sentence.

6 23: How did you derive these values of z0? While your z0 fits within the range of published values (as you discuss later in section 3.4), a reference here would be useful. Do you have different values for snow and ice or is the spatial variation for all on-glacier surfaces constant? It would be interesting to plot the snowline for this day on to Figure 1 if it is known. Are the effects of different on-glacier surfaces (snow/ice) important here, considering a constant 273.16K surface temperature?

6 28: What is the hour of the 12th August that is being reported in this paper? I think

this may be relevant for the time of day on the glacier and the expected temperature outside the glacier boundary layer and possible shading effects etc.

7 8-9: Has the size of computational domain been altered to test the resultant differences in turbulent energy generation?

7 9: What is meant by opposite DEM boundaries? I think that a new figure providing a schematic of the layers/grids used for the LES would be very useful, albeit selective of the key things to include. The description of the LES model is detailed well, though considering it comprises a large proportion of the paper, the addition of a figure could be beneficial to aid the reader.

7 15: Remove "very"

7 18: Remove "it turns out that" and add a supporting reference for M-O application.

8 14: Why these sites? Please add some brief justification/description.

8 15-16: Remove "Apparently" – Spelling mistake "luv" – Assumed to be "lee"?

8 25: Replace with "Generally, katabatic winds. . .."

9 7: "for mountain glaciers during CLEAR sky conditions".

9 12: "Similarly, . . .."

9 12-14: The downslope winds at Z4 would also be weaker due to a minimal fetch of the boundary layer too.

9 16: Please add the wind direction cases to Figure 3 as they are currently just interpreted from the same positioning as Figure 2. Also, it would be beneficial to add letters a-d to all subplots to more easily direct the reader to the appropriate information from the text.

9 16-17: This doesn't appear to be the case for the bottom left figure, which I assume to be the Northerly wind case. Are the authors only referring to the westerly (upper left)

case here?

6 15-20: I think this paragraph could do with greater clarification about which cases are being described. Again, some detail about conditions during the considered time period would be interesting. Does the free-air meteorology represent the typical cycle of the region?

10 2: Change "shapening" to " ,shaping".

10 15: Rewrite as "More importantly, the distortion. . .."

11 1: Rewrite as "On the one hand, distributed mass. . .."

11 5-6: spelling correction "of course".

11 18: I think adding Brock et al. (2006) here would be suitable.

11 31: remove "used".

13 1: Again, I think some justification for these two 'virtual' points is needed.

13 2: Change the acronyms here and elsewhere in the manuscript as Z0 and z0 (roughness) are too similar.

13 20: It is not clear where in Table 2 that 7 Wm -2 is derived from. Please clarify. Is this underestimated relative to the LES for just the west case, 6.9 Wm-2?

13 26-28: To my understanding, Figure 9 shows the differences in sensible heat fluxes between the LES and bulk method when data are extrapolated using lapse rates (Table 3) between different site combinations. It is not clear however whether a particular wind case (of the LES) is presented in the figure. As mentioned earlier, the naming convention and the way in which it changes between subsections of the paper is confusing and needs changing. Furthermore, although the test of lateral sites is interesting and an important aspect of glacier micro-meteorology to consider, why was site Zb selected in its current position? Was this randomised?

13 29-30: Re-word "lack to reflect"

13 30: You mention variability in time. However, this paper is only demonstrating statistics for one hour (p6, l27-28). Although it is likely that the bulk approach would poorly represent this temporal variability, Figure 9 does not show it.

13 32: Refer to Table 3 here.

14 1: "Similarly, . . .."

14 1: I think it is better to refer to a "shallow" temperature gradient/lapse rate rather than "small", however, the scientific community does not always agree on this and it is a minor point.

14 4-5: This is a crucial point, though it could perhaps be supported with measured data as well, which will still represent relative temperature differences at two on-glacier locations (through use of lapse rates) even if the LES isn't designed here to represent the observed absolute values.

14 7: replace "what generates" with "that generates".

14 12: Perhaps re-word this as we are talking about a much small period of time than just a summer.

14 16: Check the consistency of spelling using British/American English – here referring to "Parametrised" - ( http://www.the-cryosphere.net/for_authors/manuscript_preparation.html). (See p11, l15 / p12 l9 etc)

14 24-25: The difference in lapse rate between Z0-Za and Z0-Zc is strong, presumably due to the heat advection from the south west ridge of Zufallferner (Box R1). I think it would be useful to refer explicitly to this potentially large difference over a small (200 m?) distance on the glacier.

Cited literature

Brock BW, Willis IC and Sharp MJ (2006) Measurement and parameterization of aerodynamic roughness length variations at Haut Glacier d'Arolla , Switzerland. J. Glaciol. 52(177), 281-297

Immerzeel WW, Petersen L, Ragettli S and Pellicciotti F (2014) The importance of observed gradients of air temperature and precipitation for modeling runoff from a glacierized watershed. Water Resour. Res. 50, 2212–2226 (doi:10.1002/2013WR014506)

Shaw T, Brock B, Fyffe C, Pellicciotti F, Rutter N and Diotri F (2016) Air temperature distribution and energy balance modelling of a debris-covered glacier. J. Glaciol. 62 (231) 185-198 (doi:10.1017//jog.2016.31)
* * *

---

## Referee Comment (RC2) · Anonymous Referee #2 · 18 Jul 2016

**General comments**

In this paper the authors make use of high resolution Large-Eddy Simulations (LES), considered as a pseudo-reality testbed, to evaluate the spatial variability of the sensible heat flux over a glaciated area in the Italian Alps. The output of LES has been used for assessing the impact of using the pseudo-observations at few sites placed on the glaciers for extrapolating wind and temperature by means of linear extrapolations. The authors conclude that the magnitudes of the surface (sensible) heat flux errors are strong enough to significantly affect the surface energy balance and derived climate sensitivities of mountain glaciers.

Although rather well written and with clear figures/tables, in my opinion the paper has

several weak points and does not provide sufficient evidence in support of the authors' conclusions.

A still-open key scientific question is highlighted, i.e. how the assumptions generally made for extrapolating meteorological forcing field from sparse point observations impact the estimated local and glacier-wide melting rates. In particular, the focus is on calculation errors of the sensible heat flux distribution. However, the authors quantify this impact only comparing sensible heat flux calculations, whereas it should be assessed in comparison with the overall energy and mass balance (or melt rates). In addition, due to computational restrictions, they only perform calculations for one hour on a clear-sky day in summer 2013. I suggest evaluating the impact of sensible heat flux calculations vs. the surface energy balance, in different meteorological conditions. Moreover, I'm wondering if the mass balance measurements on Langenferner (http://acinn.uibk.ac.at/research/ice-and-climate/projects/langenferner ) could be used for estimating the impact on local and glacier-wide melt rates.

The authors claim that 'the pseudo-reality atmosphere is not required to be an observed real world case, but needs to be plausible in the sense that relevant processes are realistically simulated'. It is unclear what is meant with relevant processes. In section 4 the authors say that sections 3.1, 3.2 and 3.3 demonstrate that LES capture these relevant processes, but in these sections there is only a description of model results (some of them are obvious) and complete absence of comparison with real-world observations. In my understanding, the plausibility and realism of LES is only assessed based on the authors' personal knowledge of the atmospheric circulation over mountainous terrain, but I'm not sure that it is sufficient. On the other hand, Figure 1 shows several weather stations in the study area. Why not using these data for checking the realism of calculations? How can it be assessed that LES is superior to the bulk approach, without any comparison with real-world observations?

There is confusion between point-site process understanding and interpolated/extrapolated input meteorological fields from sparse meteorological observations

coming from on-glacier sites. If it's true and obvious that process understanding at individual sites is not sufficient to fully characterise the micrometeorological conditions over glacier surfaces, the practical or operational need to achieve such full characterization remain questionable (and in any case is not quantified in this paper).

Interpolation/extrapolation of meteorological data from on-glacier sites has limited practical usefulness. In operational model applications, there are almost no input data coming from inside the glaciers. In particular, I refer to applications aimed at exploring the climate sensitivity of glaciers, which is mentioned by the authors. Because the climatic sensitivity can be defined as 'the ratio of changes in the 2 m temperature above a glacier to changes in the temperature outside the thermal regime of that glacier (Greuell and Böhm, 1998), there is little usefulness in testing the errors coming from interpolation/extrapolation of pseudo-observed (or better, calculated) wind and temperature coming from points located inside the glaciers. It could be more useful to test calculation schemes recently proposed in the literature (cited by the authors) starting from off-glacier weather stations.

Finally, I would recommend this work for publication in The Cryosphere only after major revision and addressing of the main points reported here and in the following specific comments.

Specific comments

Page 1 line 5 and 7: please add the percentage in under-overestimations, and also the percent error in mass balance calculations. Small-scale heat flux, glacier heat fluxes... please be consistent throughout the paper and try to use always the same wording (i.e. sensible heat flux)

Page 1 line 8 and 9: it is unclear if site selection and flow direction refer to data measurements, extrapolations, or validations

Page 1 line 9-11: this is not adequately quantified in the paper. The magnitude of

sensible heat flux calculation errors and their impact on the surface energy balance and on the derived climate sensitivities should be calculated and several numbers should be added also here in the abstract.

Page 2 line 10: consider replacing 'can make over' with 'can represent'

Page 2 line 13: consider removing 'peculiar'. In this period it is partly unclear to which mass balance studies dealing with small scale variations of melt rates the authors are referring to

Page 2 line 16: what is meant exactly with 'the deficiency of monitoring activities'?

Page 2 line 26: an open scientific question

Page 2 line 27: be fully answered

Page 2 line 28: I suggest to state more clearly the aim(s) of the study

Page 3 line 19-20: SBL, SGS, please define acronyms

Page 4 line 9-10: it has been recognized

Page 4 line 13-14: the negligibility of assumptions should be demonstrated and/or possible errors coming from assumptions should be quantified

Page 4 line 19-22 and page 6 line 14-16: please see the previous comment on assumptions

Page 6 line 23: I suggest adding some references for aerodynamic roughness heights

Page 6 line 25: ERA-Interim reanalysis data (also p5 l21)

Page 6 line 27-28: from which hour to which hour of the day?

Page 7 line 9-12: this part is somewhat unclear and it looks like the authors adjust the DEM (the only real-world component in this work) to the requirements of the numerical model. Is it correct? Please see comment to Page 4 line 13-14

Page 9 line 4: please reword 'the intensity of the cross-valley circulation' to improve clarity

Page 9 line 23-24: consider replacing 'do not jointly appear with high wind velocities' with something like 'do not appear in the areas with high wind velocities'

Page 10 line 34: can you quantify (or estimate) the percent contribution of the sensible heat flux to the total energy balance in your case study? This would be important for understanding the impact of calculated sensible heat flux on local-scale and area-averaged energy and mass balance

Page 11 line 10-11: can you provide some numbers in support to this statement?

Section 3.4: in Figure 1 two weather stations are shown on the glaciers. Why data coming from these weather stations were not used for checking the reliability of LES experiments?

Page 11 line 19-23: with the authors, I recognize that this is a strong assumption, in particular over glaciers with such high range of elevation (2595-3750 m), quite different from the end-of-summer situation reported for Arolla by Brock et al., (2000). It should be possible to map the snow cover for the selected day, or to use another day with available snow cover data (e.g. from Landsat imagery). Alternatively, the authors should at least quantify the possible errors stemming from this assumption.

Page 12 line 2-3: on which bases the authors say that the SGS model 'seems to work well' in their study?

Page 12 line 4: maybe reword the title as 'Estimation of the sensible heat using the Bulk-Approach'

Page 12 line 20: replace 'given that' with 'in case' (I guess that it is meant where there is a weather station measuring the required variables)

Page 12 line 20-27: please consider moving this part in the following section

Page 12 line 26-27: this is a strong statement, because there is complete absence of comparison between modelled and observed (relevant) processes. What are relevant processes? How can the authors assess that LES captures observations, without reporting observations or without citing literature on this topic?

Page 13 line 3: please replace 'pseudo-observed' with 'calculated'. I guess these are temperature and wind speed data calculated using LES, is it right? Please specify

Page 13 line 5: surface heat flux, surface sensible flux, or surface sensible heat flux? Please be consistent

Page 13 line 7-9: please explain why there are differences at the two Za and Z0 sites, given that (in my understanding) wind speed and temperature at these sites are the same using the bulk method and the LES (i.e. they differ in the rest of the analysed area, but not at Za and Z0).

Page 13 line 20: the average sensible heat flux (please, add % error in the text). How big is the impact on glacier-wide total energy balance calculations?

Page 13 line 27: for which wind direction?

Page 13 line 28: using linear extrapolations across the glaciers?

Page 13 line 29: in my opinion there is an equivocal use of the term 'bulk method', which is a method for calculating turbulent exchanges, referred to the calculations using linear extrapolations across the glaciers. I would suggest clarify/avoid ambiguities

Page 13 line 33 and in the following: please check or clarify, if gradients are too large (in absolute value) underestimations of temperature and sensible heat flux should occur in the upper parts of the glaciers. Moreover, in absence of model validation, why the LES model has to be the right one and the Bulk has to be the wrong one, a priori?

Page 14 line 3: also in this case I suggest to calculate the relative importance of these errors in the overall energy balance of the glacier

Page 14 line 5-8: this part is methodological and should be moved at the beginning of Sect. 4.2. It also deserves rephrasing to improve clarity

Page 14 line 12: it is unclear why the authors selected only a clear-sky case study

Page 14 line 17-22: I have several points, which could/should be at least partly addressed or discussed in the manuscript. In particular they concern: i) the practical or operational need to fully characterise the micrometeorological conditions over glacier surfaces; ii) the linear extrapolation of forcing fields from sites placed over glaciers (again, almost never available in practical model applications); iii) related to the previous point, the climate sensitivity has to be assessed with respect to climatic conditions observed outside the microclimatic influence of the glaciers.

Page 14 line 20: here and elsewhere, I suggest speaking about differences and not errors, because the comparison is between calculations and not between calculations and observations

Page 14 line 30: percent error of what?

Page 15 line 1-2: when small-scale variations of surface energy balance are required? Please add this in the introduction and recall it here and/or in the abstract

Page 14 line 6: using off-glacier stations for what?

Comments on the figures and tables:

Figure 1: this image lacks east-north coordinates or inset displaying wider geographical setting of the study area. Four weather stations are reported, whose data are not used in this paper

Figure 2 (and following maps): I suggest adding some contour line (or hillshaded DTM, like in Fig. 1), which is needed for a better understanding of the local topography and of its effects on the calculated variables

Figure 9: in the caption just begin with 'differences in the surface....' and correct

pseudo-observations coherently with the text

Table 3: I suggest adding LES estimates and % differences (not error, please correct also in Table 2) as in Table 2.

---

## Author Comment (AC1) · 21 Sep 2016

We very much thank the two reviewer for their thorough analysis of our article and for their valuable comments, annotations and suggested improvements. They had been carefully considered and most of them are accounted for in the revised manuscript. Answers and explanations to all detailed questions and annotations raised by the reviewers are provided in the following.
(RC: Reviewer comments; AC: Author comments)

**RC 1:** Due to the computationally intensive nature of the LES, it is understandable that a small timeframe is most suitable to demonstrate the expected variation of sensible heat fluxes over the glaciers. However, I think the paper would benefit from having more detail on the conditions of the hour for which statistics are presented. The authors describe a blue sky condition which is known to be favourable for the development of a katabatic boundary layer, however the strength of the boundary layer can also be affected by the ambient air temperature (data from the off-glacier sites seen in Figure 1 could aid this). Furthermore, could the LES be compared with a cooler/cloudier hour? Though adding some extra work, I think this would benefit the scientific community and be informative for when (under which conditions) sensible heat fluxes are most likely to be inadequately modelled.

*AC: Measures characterizing the atmospheric condition (ambient conditions), such as lapse rates or heating rates, depend on the locations where the measurements are taken. At the slopes there is a well-mixed layer (~10-50 m) with nearly constant potential temperature (~10℃) and a thermally driven slope wind develops. The synoptic flow enhances or retards the slope winds and alters the temperature distribution. To test this, we have calculated the lapse rate on an east slope for each experiment (different large-scale forcing). When the large-scale flow aligns with the slope winds (easterly flow) the lapse rate is lower (0.0067 K/m or even lower) than for the other cases (~0.0078 K/m or even higher). We have attached two figures to this review to illustrate the advection of warm air over a ridge and how it impacts the lapse rates (ridge_east.pdf and ridge_west.pdf). The same argument holds for the heating rates of the near-surface layer. Therefore, it might be the best to provide a vertical atmospheric profile at the location Z2 on the Zufallferner. A new figure has been added showing the vertical temperature profile up to 10000 m. Above the Cevedale Peak the lapse rate is approximately -0.006 K/m, which corresponds to the profile given by the ERA-Interim data. Together with the temperature deficit (between the 2 m temperature and the free atmosphere, see Table 1) this provides a valuable information on the ambient air temperature in the valley.*

*We also like to note, that the atmospheric background state for temperature and pressure from the ERA-Interim data was from the 17th August 2014 and not 12th August 2013 as given previously in the text (p6 L25/26).*

*We agree with the reviewer that the scientific community would benefit from a greater variety of cases and more general conclusions on the sensible heat fluxes. In order to draw a general conclusion, however, a large number of experiments is needed to cover the wide spectrum of topographic and atmospheric constellations. Unfortunately, we have already reached our computational capacities and try to solve this in an upcoming project. Each LES run of 9 hours' simulation time requires a computational time of 5-7 days on a High-Performance Computer with 400 cores. This is the first time that high-resolution LES have been performed over alpine glaciers and it shows that this approach has potential to study small scale processes.*

**RC 2:** As the work details, the LES is not required to be an observed real-world case, as the realistic simulation of processes and their spatial variation is key. However, the authors indicate several weather stations in Figure 1 (which are not used). It would be interesting to present what the actual lapse rate on glacier would be and also compare the calculation of sensible heat fluxes using this measured data. If no AWS measurements are to be utilised in this study, please remove them from the figure.

*AC: In an idealized setup, the surrogate atmosphere can only be compared with well-known characteristics of stable boundary layer and dynamical atmospheric features obtained from in-situ*

*measurements on alpine glacier. These characteristics include the vertical (wind and temperature), sensible heat flux and turbulent structure of the boundary layer and should be of the same order of magnitude as the measurements (Section 3.1 and 3.2).*

*During the week of the 17th August 2014 we had temporarily installed two weather stations, one closed to Z1 on Zufallferner and another further down the valley. The glacier station measured between 13 and 14 h a mean wind velocity of 4.6 m/s at 2 m height above the surface. Even though this is closed to the simulated value (4.5-6 m/s, westerly flow), the two values are not comparable at all. The prescribed surface heating rate (1.2 K/h) of the surroundings is lower than the measured heating rate (4.1 K/hr) at that particular day. Furthermore, the idealized simulations do not account for differential heating by radiation which is important during the first two hours and leads to asymmetric cross-valley winds. Without doubt, the homogenous heating assumption is a major drawback of the code. Although the chosen heating rate is significantly lower and shadowing effects are absent the typical low level jet and the heat advection from the lateral boundaries are present. As indicated in the conclusion, due to conservative chosen boundary conditions the simulated advection effects might be weaker than the one observed in a real atmosphere.*

*To avoid confusion, we follow the recommendation of the reviewer and removed the stations from Figure 1.*

**RC 3:** The authors also outline several sub-regions and 'virtual' sites of interest on Zufallferner though with no clear justification for why. I think it is important to demonstrate the spatial variation of wind fields along a glacier centreline and focus on specific sites (i.e. Z1-Z4), particularly when attempting to simulate and understand interactions of the glacier boundary layer with synoptic scale winds. Furthermore, the selection of temperature extrapolation locations is important although often somewhat arbitrary in many studies. However, the presentation of several different sites between figures (Figures 7,8,9 for example) and their naming conventions (Z3 changes to Za then to Zc) is misleading. The authors should add some additional reasoning to their choices of virtual sites. The authors should also guide the reader to aspects of figure subplots by labelling them (i.e. a-d). Misleading information for Figure 3 is particularly noteworthy.

*AC: We thought, virtual sites make it easier for the reader to follow the discussion. Each region shows a different flow pattern: (R1) ridge region with flow separation, (R2) a steep ice fall, (R3) katabatic wind region, and (R4) divergence of katabatic winds. We have removed the sub-regions in Figure 4, 6 and 7, while we kept the regions in Figure 2 for discussion. The justification for the regions is now given in the first paragraph of Section 3.1.*

*"For the discussion we introduce four specific regions: (R1) ridge region, (R2) a steep ice fall, (R3) katabatic wind region, and (R4) divergence zone of katabatic wind. Local characteristics are discussed at four virtual sites on the glacier (Z1-4)."*

*We agree, that the focus of the discussion should be on the winds along the glacier centerline (Z1-Z4), and we think that has been done since most of the discussion of Section 3.1 is related to the wind fields on Zufallferner. However, the discussion on the dynamic and cross-slope winds (second paragraph of Section 3.1) helps to better understand the processes (interaction with the synoptic and thermal winds) that cause the spatial variation along the centerline.*

*Yes, the naming convention is misleading. We have changed labels of the locations used for interpolations to (S1-S5) and also labelled the subplots to guide the reader. From the text it is indeed not obvious how we have chosen the sites. The idea was to select sites with distinct flow and advection patterns: (Z0) at the tongue with almost pure katabatic wind (used as reference station), (Za) in the higher region which is influenced by strong advection, (Zb) at the lateral boundary of the glacier which is influenced by the cross-valley circulation, (Zc) very closed to Za but not affected by strong*

*heat advection, and (Zd) a second station on the glacier with dominantly katabatic wind. We now give the reason to our choice in the second paragraph of Section 4.2.*

*"To explore how the choice of observation sites influences the spatial variation of the surface heat flux estimates, we define a set of virtual observation on Zufallferner with distinct flow and advection patterns: (S1) located at the glacier tongue with almost pure katabatic wind (used as reference station), (S2) in the higher region which is influenced by strong heat advection, (S3) at the lateral boundary of the glacier which is influenced by the cross-valley circulation, (S4) closed to S2 but less affected by strong heat advection, and (S5) a second station on the glacier with dominantly katabatic wind. For each combination of S1 and S2-S5 the heat fluxes are estimated according to Eq. 16."*

**RC 4:** Finally, while it is clear from section 1 what the problems of the literature are (and it is very well written), I think it is important to stress in a little more detail what the aim of the paper is and add some more discussion regarding the applicability of an LES approach at the end.

> *AC: In the last two paragraphs of the introduction we now stress in more detail the aim of the paper.*
>
> *"To overcome this difficulty, we make use of high resolution Large-Eddy Simulations (LES). The LES are considered as pseudo-reality - a testbed to identify the shortcomings in the local surface heat flux estimates when the lack of observations restrict our micrometeorological knowledge to a few sites. The plausibility of the temperature interpolation algorithms and the derived surface heat fluxes can be more strictly tested in a surrogate world of atmospheric simulations, which offer a realization of atmospheric states in which all target variables are known. The pseudo-reality atmosphere is not required to be an observed real world case, but needs to be plausible realization of the atmosphere in the sense that relevant processes are realistically simulated. The advantage of such studies is that the surrogate atmosphere provides a perfect pseudo-observation of all the variables required to establish the skill of an interpolation method and hence the surface heat flux calculations. While surrogate atmospheres have been widely used in downscaling studies it's still a new approach in glaciological studies (Frias et al., 2006; Vrac et al., 2007; Maraun, 2012)."*
>
> *Frías, M. D., E. Zorita, J. Fernández, and C. Rodríguez-Puebla (2006), Testing statistical downscaling methods in simulated climates, Geophys. Res. Lett., 33, L19807, doi:10.1029/2006GL027453.*
>
> *Vrac, M., M. L. Stein, K. Hayhoe, and X.-Z. Liang (2007), A general method for validating statistical downscaling methods under future climate change, Geophys. Res. Lett., 34, L18701.*
>
> *Maraun, D. (2012), Nonstationarities of regional climate model biases in European seasonal mean temperature and precipitation sums, Geophys. Res. Lett., 39, L06706, doi:10.1029/2012GL051210.*

Specific comments

**RC:** 1 7: Add the temporal scale for which of the flux over- and under-estimates are found (i.e. 1 hour of statistics).

> *AC: Changed to: "The glacier-wide hourly averaged surface heat fluxes are both over- and underestimated by up to 16 $Wm^{-2}$ when using extrapolated temperature and wind fields."*

**RC:** 1 18: Re-word "loss of information".

**AC:** *Changed to: "The reduced spatial and temporal variability ..."*

**RC:** 2 10: I think it is important to stress that this "over 50%" contribution from turbulent heat fluxes is typical for overcast conditions or for maritime glaciers (as is given by the studies you cite –e.g. Cullen and Conway, 2015) as otherwise the dominance is typically, from shortwave radiation. For your study you assess a clear sky condition and a continental glacier.

> **AC:** *We have re-written that sentence: "The energy surplus can be critical for the ablation, considering that the turbulent heat flux can represent 50% of the total energy during pronounced melt events on maritime mid-latitude mountain glaciers in summer, and even up to 30% on continental glaciers (e.g. Cullen and Conway, 2015; Gillett and Cullen, 2011; Van den Broeke, 1997; Hock, 2005; Klok and Oerlemans, 2002; Oerlemans and Klok, 2002; Giessen et al., 2008; Moore and Owens, 1984)."*

**RC:** 2 13: Replace "peculiar" with "particular".

> **AC:** *Done.*

**RC:** 2 25-26: Though I agree that there is still much to be understood about the impact of these assumptions on glacier melt rates, citing some of the work which has made attempts to use distributed temperature for this purpose would be suitable here. For example, Immerzeel et al. (2014) investigate this for a catchment/valley scale and Shaw et al. (2016) investigate this for a debris-covered glacier.

> **AC:** *Done.*

**RC:** 3 19: I assume here that you refer to the surface boundary layer for "SBL"? Write out in full before using the acronym.

> **AC:** *Yes, SBL refers to stable boundary layer. Done.*

**RC:** 3 20: What does SGS refer to? Write out in full as well.

> **AC:** *SGS refers to subgrid-scale. Done.*

**RC:** 4 4: A minor point, but you are missing an equation number for eddy viscosity (this should be eqn 7).

> **AC:** *We have added the missing equation number.*

**RC:** 4 9-11: This sentence needs re-writing. It is unclear what it is trying to say and the

sentence has syntax errors.

> *AC: The sentence has changed to: "While, energy is transferred from the large to small scales according to the Kolmogorov energy cascade, it has been observed that locally there can be a significant transfer of energy from the residual motions to the resolved scales (backscatter)."*

**RC:** 5 2: Changes in temperature and phase from radiative forcing would be relevant if the LES approach was adopted over a longer time-frame. This may be worth adding to the discussion?

> *AC: In the last paragraph of Section 3.4 we now indicate the how insolation on slopes affect the circulation pattern.*
>
> *"We like to note, that the current version of the solver ignores differential surface heating by radiation and is therefore only suitable for idealized simulations. Differences in insolation on slopes due to exposure, aspect or shadow cause upslope flows to be inhomogeneous. The different onsets of the slope winds then lead to more asymmetric cross-valley circulations."*

**RC:** 5 16-17: How is the topography representative of many in the European Alps? Can you also add the mean slope of the glacier to this section?

> *AC: We have added more topographic information to this section:*
>
> *"The surface area of the glaciers is about 6.62 km² (2013) with an altitudinal extent from about 3750 m a.s.l near the summit of Hintere Zufallspitze, down to 2595 m a.s.l. at the lowest point of Zufallferner. The model domain includes a wide variety of topographic features such as steep slopes up to 50°, glaciated and unglaciated (summit-) ridges of various aspects, as well as larger glacier sections with smooth terrain and low slope angles. The mean slope angle of the glacierized terrain is 17°. The topography can be regarded as (i) typical for many glaciers in the European Alps and (ii) highly suitable for investigating the complex interaction of large-scale (synoptic) forcing and small scale topographic features."*

**RC:** 5 21: What grid size do you use for the ERA-Interim reanalysis data? Is this re-sampled from the 6 hourly temporal scale of ERA-Interim? Additional detail would be useful here.

> *AC: The ERA-Interim reanalysis data is available on a 0.75x075 degree grid. The ERA grid cell data above the investigation is mapped onto the LES grid. We have initialized the LES model with the vertical profile from 06UTC. It now reads:*
>
> *"The atmospheric background state for temperature and pressure is derived from ERA-Interim reanalysis data from 06 UTC. The vertical data is uniformly mapped onto the unstructured LES grid."*

**RC:** 5 28: Specify if the 100 m temperature is that from the ERA-Interim.

> *AC: Yes, the 100 m temperature is that from the ERA-Interim data. We have included now this information: "The pre-factor, C, is the temperature perturbation at the glacier surface, which in our case is the difference between surface temperature (273.16 K) and the ERA-Interim*

*temperature at 100 m above the surface."*

**RC:** 6 2: Why 8 m/s-1? Is this the mean value from the given six hour period of the reanalysis data?

> *AC: Yes, this is the mean wind velocity from the ERA-Interim data at 5500 m. We have added the following sentence at the end of the paragraph: "This corresponds to the mean wind velocity of the ERA-Interim data at 5500 m."*

**RC:** 6 8: It is unclear what you mean by this - "some sort of model". Please re-word this sentence.

> *AC: We have re-worded this phrase: "The filter and grid resolution are too coarse to resolve the near-wall motions, including in the viscous wall region, so that their influence closed to the wall are modelled by a shear stress model."*

**RC:** 6 23: How did you derive these values of z0? While your z0 fits within the range of published values (as you discuss later in section 3.4), a reference here would be useful. Do you have different values for snow and ice or is the spatial variation for all on-glacier surfaces constant? It would be interesting to plot the snowline for this day on to Figure 1 if it is known. Are the effects of different on-glacier surfaces (snow/ice) important here, considering a constant 273.16K surface temperature?

> *AC: The values have been taken from literature. We have included some references.*
>
> *The roughness length for snow and ice are the same. We have added the following sentence: "The aerodynamic roughness height, z0, is set to 0.1 m for the land surfaces (e.g. Stull, 2012) and to 0.001 m for the glacier and snow surface (e.g. Braithwaite, 1995; Giessen et al., 2008; Brock et al., 2000; Hock, 2005; Greuell and Smeets, 2001), respectively. We assume similar roughness height for snow and ice since large parts of the glaciers were covered by a thin layer of fresh snow."*
>
> *This assumption is also discussed in Section 3.4:*
> *"A crucial assumption is the surface roughness length. To obtain more general results, uniform values of z0 for snow and ice with 0.001 m are used, which is in the range of commonly used values (e.g. Braithwaite, 1995; Giessen et al., 2008; Brock et al., 2000; Hock, 2005; Greuell and Smeets, 2001). The 'uniform' assumption ignores temporal and spatial roughness length variations. However, potentially such variations can have a strong influence on the magnitude of the surface energy fluxes (Brock et al., 2000; Giessen et al., 2008). We argue that this assumption is acceptable since large parts of the glaciers were covered by a thin layer of fresh snow."*
>
> *I think you refer to the effects of the surface characteristic on the atmosphere. Different roughness height would certainly impact the momentum flux and heat exchange at the surface. However, we think that it is more important (at least in the summer season) to account for non-uniform roughness changes, e.g. seracs, ice falls or the sudden change in roughness at the glacier boundary. While elements such as seracs are not resolved the model accounts for the sudden roughness changes at the glacier boundary. On large glaciers (e.g. Kronebreen and Kongsvegen) the sudden roughness change at the tongue due to huge seracs has severe effects on the flow. The Zufallferner is rather small and the influence from the surrounding may*

*overwhelm the errors made by this assumption.*

**RC:** 6 28: What is the hour of the 12th August that is being reported in this paper? I think this may be relevant for the time of day on the glacier and the expected temperature outside the glacier boundary layer and possible shading effects etc.

> *AC: The model has been initialized with the ERA-Interim profile from 06 UTC (see comment above) and a uniform surface temperature of 273.16 K (Section 2.3). On p6L28 we refer to the last simulation hour. We have now added this information.*
>
> *As mentioned in the second comment, the idealized simulations do not account for differential heating by radiation (shading effect). The surface temperature of the surrounding is given by the prescribed surface heating rate (1.2 K/h). At the end of the simulation the surface temperature is 10.8 K.*

**RC:** 7 8-9: Has the size of computational domain been altered to test the resultant differences in turbulent energy generation?

> *AC: Yes, we have tested various simulation setting. One concern was the development of gravity waves which would impact the boundary layer characteristics. However, we could not find significant differences between a domain size of ~15 km and ~10 km (and 12.5 m horizontal resolution). The simulations are more sensitive to the choice of the grid size. Only 60-70% of the kinetic energy was resolved when using a horizontal resolution of 25 m. Additionally, decreasing the horizontal resolution lead to greater aspect ratios of the prismatic layers, which required even shorter integration time steps (0.01 s). Decreasing the prismatic layers was not an option since this would affect the shear stress and momentum calculations closed to the surface. The choice of ~12 m was a good tradeoff between computational costs and model quality. Besides the computational domain setup, the choice of the subgrid-scale model is essential for the results. The Smagorinsky SGS model was to dissipative in the stable boundary layer which led to numerical instabilities.*
>
> *We have added the following text to Section 3.4: "When decreasing the horizontal grid resolution to 25 m the resolved kinetic energy was only 60-70%. Additionally, a coarser grid leads to greater aspect ratios of the prismatic layers, which requires very short integration time steps (0.01 s) to guarantee stability. Increasing the prismatic layer heights is problematic since this affects the shear stress and momentum calculations closed to the surface. The choice of ~12.5 m is a good tradeoff between computational costs and resolved scales."*
>
> *"We have also tested the dynamic Smagorinsky model, but the simulations are found to be unstable due to large fluctuations of $C_s$."*
>
> *Additionally, we have added a new paragraph at the end of Section 3.4 which should highlight the limitation of the LES solver: "We like to note, that the current version of the solver ignores differential surface heating by radiation and is therefore only suitable for idealized simulations. Differences in insolation on slopes due to exposure, aspect or shadow cause upslope flows to be inhomogeneous. The different onsets of the slope winds then lead to more asymmetric cross-valley circulations."*

**RC:** 7 9: What is meant by opposite DEM boundaries? I think that a new figure providing a schematic of the layers/grids used for the LES would be very useful, albeit selective

of the key things to include. The description of the LES model is detailed well, though considering it comprises a large proportion of the paper, the addition of a figure could be beneficial to aid the reader.

> *AC: In order to guarantee a fully turbulent atmosphere the boundaries are specified as period. Such boundaries require that faces on the opposite boundary (faces of grid cells) are equal within a certain tolerance. To do so the mesh grid points on opposite boundaries have been slowly displaced to match each other. The inner grid points are relaxed to get a smooth transition from the boundaries towards the inner domain. We have added a new figure showing a sketch of the relaxation procedure.*

**RC:** 7 15: Remove "very"

> *AC: Done.*

**RC:** 7 18: Remove "it turns out that" and add a supporting reference for M-O application.

> *AC: Done.*

**RC:** 8 14: Why these sites? Please add some brief justification/description.

> *AC: We've added a justification for that choice (see comment above).*

**RC:** 8 15-16: Remove "Apparently" – Spelling mistake "luv" – Assumed to be "lee"?

> *AC: Done.*

**RC:** 8 25: Replace with "Generally, katabatic winds: : :."

> *AC: Done.*

**RC:** 9 7: "for mountain glaciers during CLEAR sky conditions".

> *AC: Done.*

**RC:** 9 12: "Similarly, : : :."

> *AC: Done.*

**RC:** 9 12-14: The downslope winds at Z4 would also be weaker due to a minimal fetch of

the boundary layer too.

> *AC: Yes, this is an important aspect which we have included now: "Similarly, a reduced fetch and, in particular, a strong shear associated with a rapid veering of the winds with height can drastically reduce the wind velocity."*

**RC:** 9 16: Please add the wind direction cases to Figure 3 as they are currently just interpreted from the same positioning as Figure 2. Also, it would be beneficial to add letters a-d to all subplots to more easily direct the reader to the appropriate information from the text.

> *AC: Done (see comment above).*

**RC:** 9 16-17: This doesn't appear to be the case for the bottom left figure, which I assume to be the Northerly wind case. Are the authors only referring to the westerly (upper left) case here?

> *AC: We have added a comment to which Figure and subplots we are referring to.*
>
> *"The intensity and height of the wind maximum decreases down-slope for most cases (see Fig. 5a, b, d), …"*

**RC:** 9 15-20: I think this paragraph could do with greater clarification about which cases are being described. Again, some detail about conditions during the considered time period would be interesting. Does the free-air meteorology represent the typical cycle of the region?

> *AC: We now refer to the specific cases and have given more details on the ambient conditions (see comment 1). The free-air meteorology indeed represents a typical stratification for the region (see Figure 4).*

**RC:** 10 2: Change "shapening" to " ,shaping".

> *AC: Done.*

**RC:** 10 15: Rewrite as "More importantly, the distortion: : :.."

> *AC: Done.*

**RC:** 11 1: Rewrite as "On the one hand, distributed mass: : :."

> *AC: Done.*

**RC:** 11 5-6: spelling correction "of course".

 *AC: Done.*

**RC:** 11 18: I think adding Brock et al. (2006) here would be suitable.

 *AC: Yes, this reference absolutely suits here and has been added.*

**RC:** 11 31: remove "used".

 *AC: Done.*

**RC:** 13 1: Again, I think some justification for these two 'virtual' points is needed.

 *AC: To test the influence of the flow direction on the lapse rates and derived surface heat fluxes the location were chosen in a way to have a preferable large vertical altitude difference between the stations. We have given this justification in text: "To illustrate how the flux estimates depend on the local flow conditions, we defined two virtual observation points at Zufallferner, with preferable great vertical altitude differences between the sites (S1 and S2, see Fig. 10)."*

**RC:** 13 2: Change the acronyms here and elsewhere in the manuscript as Z0 and z0 (roughness) are too similar.

 *AC: We have changed Z0 to S1.*

**RC:** 13 20: It is not clear where in Table 2 that 7 Wm -2 is derived from. Please clarify. Is this underestimated relative to the LES for just the west case, 6.9 Wm-2?

 *AC: We have rewritten this paragraph:*

 *"On a glacier-scale, the bulk approach underestimates the average heat flux between 5.2 (-16.6%) and 6.9 $Wm^{-2}$ (-20.3%) for the westerly, easterly and northerly flow (see Tab. 2). The local differences for the southerly case, however, almost cancel each other (0.8 $Wm^{-2}$, 2.2%) ."*

**RC:** 13 26-28: To my understanding, Figure 9 shows the differences in sensible heat fluxes between the LES and bulk method when data are extrapolated using lapse rates (Table 3) between different site combinations. It is not clear however whether a particular wind case (of the LES) is presented in the figure. As mentioned earlier, the naming convention and the way in which it changes between subsections of the paper is confusing and needs changing. Furthermore, although the test of lateral sites is interesting and an important aspect of glacier micro-meteorology to consider, why was site Zb selected in its current position? Was this randomised?

*AC: Yes, Fig. 9 shows the differences in sensible heat fluxes between LES and bulk method using the westerly flow case. The site (Zb, now called S3) is located at the boundary of the glacier which is influenced by the cross-valley circulation. We have now given a justification of the choice (see comment above):*

*"To explore how the choice of observation sites influences the spatial variation of the surface heat flux estimates, we define a set of virtual observation on Zufallferner with distinct flow and advection patterns: (S1) located at the glacier tongue with almost pure katabatic wind (used as reference station), (S2) in the higher region which is influenced by strong heat advection, (S3) at the lateral boundary of the glacier which is influenced by the cross-valley circulation, (S4) closed to S2 but less affected by strong heat advection, and (S5) a second station on the glacier with dominantly katabatic wind. For each combination of S1 and S2-S5 the heat fluxes are estimated according to Eq. 16."*

**RC:** 13 29-30: Re-word "lack to reflect"

*AC: We have changed the sentence to: "Evidently, the bulk approach in concert with interpolated temperature fields underestimates the spatial surface heat flux variability."*

**RC:** 13 30: You mention variability in time. However, this paper is only demonstrating statistics for one hour (p6, l27-28). Although it is likely that the bulk approach would poorly represent this temporal variability, Figure 9 does not show it.

*AC: That's correct. We have removed the comment on the temporal variability (see comment above).*

**RC:** 13 32: Refer to Table 3 here.

*AC: Done.*

**RC:** 14 1: "Similarly, : : :."

*AC: Changed.*

**RC:** 14 1: I think it is better to refer to a "shallow" temperature gradient/lapse rate rather than "small", however, the scientific community does not always agree on this and it is a minor point.

*AC: We have followed your recommendation and used the expression 'shallow'.*

**RC:** 14 4-5: This is a crucial point, though it could perhaps be supported with measured data as well, which will still represent relative temperature differences at two on-glacier locations (through use of lapse rates) even if the LES isn't designed here to represent the observed absolute values.

**AC:** Please refer to RC 2, where we have discussed this issue.

**RC:** 14 7: replace "what generates" with "that generates".

>*AC: Changed.*

**RC:** 14 12: Perhaps re-word this as we are talking about a much small period of time than just a summer.

>*AC: We have re-written the sentence as follows: "The idealized LES experiments demonstrate that heat advection associated with the wind systems shape the thermal conditions on the glaciers during the course of a summer day with clear sky conditions."*

**RC:** 14 16: Check the consistency of spelling using British/American English – here referring to "Parametrised" - ( http://www.thecryosphere. net/for_authors/manuscript_preparation.html). (See p11, l15 / p12 l9 etc)

>*AC: We have checked the consistency of spelling.*

**RC:** 14 24-25: The difference in lapse rate between Z0-Za and Z0-Zc is strong, presumably due to the heat advection from the south west ridge of Zufallferner (Box R1). I think it would be useful to refer explicitly to this potentially large difference over a small (200 m?) distance on the glacier.

>*AC: We have taken up this idea and added the following sentences: "Generally, the sensitivity of the calculated lapse rates to the choice of the observation sites is related to the steep gradients between the advected warm air masses and the ambient cold air masses on the glacier. Shifting stations by even small distances (<= 200 m) can potentially lead to remarkable differences in the calculated lapse rates of $\pm 0.005\ Km^{-1}$."*

---

## Author Comment (AC2) · 21 Sep 2016

We very much thank the two reviewer for their thorough analysis of our article and for their valuable comments, annotations and suggested improvements. They had been carefully considered and most of them are accounted for in the revised manuscript. Answers and explanations to all detailed questions and annotations raised by the reviewers are provided in the following.
(RC: Reviewer comments; AC: Author comments)

**Comment 1**) A still-open key scientific question is highlighted, i.e. how the assumptions generally made for extrapolating meteorological forcing field from sparse point observations impact the estimated local and glacier-wide melting rates. In particular, the focus is on calculation errors of the sensible heat flux distribution. However, the authors quantify this impact only comparing sensible heat flux calculations, whereas it should be assessed in comparison with the overall energy and mass balance (or melt rates).

*AC: The study focuses on the effect of local advection on the spatial sensible heat flux variation on glaciers and to test the skill of commonly used approaches to estimate the surface heat fluxes at a given point on the glacier. While, without doubt, the impact of the heat flux variation on the glacier mass balance is of major interest, we have focused on the sensible heat flux for the following reasons:*

*(i)     Many scientific studies have revealed that especially for mid latitude mountain glaciers, the sensible heat flux, after the net radiation budget, constitutes the main energy source and consequently explains a large part of observed glacier ablation (e.g. Braithwaite, 1995; Smeets et al., 1998; Oerlemans, 2010; Gillett and Cullen, 2011; Senese et al., 2012; Conway and Cullen, 2013; Cullen and Conway, 2015). The emphasis of most studies is placed on the averaged turbulence conditions at a given point over glaciers and their impact on the surface energy balance. These studies achieved significant progress by generalizing results with respect to the inherent physical processes or mechanisms at a point scale. The spatial variability of the turbulent quantities, however, has received much less attention than the time averaged quantities.*

*(ii)    As mentioned in the introduction, the complex interaction of glaciers, atmosphere and topography constitutes a fundamental challenge to environmental research. Non-local topographic effects control the micrometeorological conditions on glaciers, but the process itself is challenging to study. In order to reduce the degree of freedom, we exclude all quantities in the idealized simulations which are not directly affected by the flow, but are known to be important for the surface energy balance e.g. radiation divergence, conservation of moisture.*

*(iii)   To study the impacts of the sensible heat flux on the overall energy and mass balance require a direct coupling (online) of the LES with a mass balance module, which we are currently implementing in the LES solver. However, this module introduces additional initial/boundary conditions and requires rather long spin-up times. Without well-posed boundary and initial conditions (e.g. soil properties or moisture), the problem gains complexity and adds additional degrees of freedom.*

*(iv)    The research goal was already very ambitious. There very few studies dealing with LES in (very) complex terrain and in particular over glaciers. However, this study illustrates that there is a potential in studying the surface energy and mass balance on mountain glacier. If LES are useful for real case studies is yet to be answered.*

**Comment 2**) In addition, due to computational restrictions, they only perform calculations for one hour on a clear-sky day in summer 2013. I suggest evaluating the impact of sensible heat flux calculations vs. the surface energy balance, in different meteorological conditions. Moreover, I'm wondering if the mass balance measurements on Langenferner (http://acinn.uibk.ac.at/research/ice-and-climate/projects/langenferner) could be used for estimating the impact on local and glacier-wide melt rates.

*AC: We agree that the contribution of the sensible heat flux to the surface energy balance is important to understand the impacts on the mass-balance. However, the focus of this study is the*

*effect of local advection on the spatial sensible heat flux variation on glaciers and to test the skill of commonly used approaches to estimate the surface heat fluxes at a given point on the glacier (see also Comment 1). In order to understand the impact, the LES (including radiation) must be coupled directly with a distributed mass balance model and integrated over longer time periods. We have re-written the introduction to emphasize our research goals (see Comment 3).*

*In order to draw a general conclusion (not only for clear-sky), however, a large number of experiments is needed to cover the wide spectrum of topographic and atmospheric constellations. Unfortunately, we have already reached our computational capacities and try to solve this in an upcoming project. Each LES run of 9 hours' simulation time requires a computational time of 5-7 days on a High-Performance Computer with 400 cores. For that reason, we have focused on a clear-sky case of which we have expected pronounced thermal wind phenomena and heat advection. The latter one is an important process to understand the thermal conditions on glaciers.*

*The timescale of our simulations is a few hours, while the scale of mass balance measurements and stake readings is several weeks to months. Consequently, the direct measurements cannot be used for any impact assessments. Our study is motivated by the findings of many previous studies which prove the general importance of the sensible heat flux for mid latitude glacier melt (e.g. Klok and Oerlemans, 2002; Oerlemans, 2010; Gillett and Cullen, 2011; Senese et al., 2012; Conway and Cullen, 2013; Cullen and Conway, 2015).*

**Comment 3)** The authors claim that 'the pseudo-reality atmosphere is not required to be an observed real world case, but needs to be plausible in the sense that relevant processes are realistically simulated'. It is unclear what is meant with relevant processes. In section 4 the authors say that sections 3.1, 3.2 and 3.3 demonstrate that LES capture these relevant processes, but in these sections there is only a description of model results (some of them are obvious) and complete absence of comparison with real-world observations. In my understanding, the plausibility and realism of LES is only assessed based on the authors' personal knowledge of the atmospheric circulation over mountainous terrain, but I'm not sure that it is sufficient. On the other hand, Figure 1 shows several weather stations in the study area. Why not using these data for checking the realism of calculations? How can it be assessed that LES is superior to the bulk approach, without any comparison with real-world observations?

*AC: In an idealized setup, the surrogate atmosphere can only be compared with well-known characteristics of boundary layers and dynamical atmospheric features obtained from in-situ measurements on alpine glacier. These characteristics include the vertical (wind and temperature), sensible heat flux and turbulent structure of the boundary layer and should be of the same order of magnitude as the measurements. In Section 3.1 and 3.2 we compare the wind magnitude, LLJ, intermittency and turbulence scales with observation made by other studies. For example:*

*i)    The wind magnitudes are characteristic for mountain glacier during clear sky conditions (e.g. Van den Broeke, 1997; Söderberg and Parmhed, 2006).*

*ii)   Several studies observed intermittent turbulent mixing events in the SBL above glaciers and analyzed their impact on the surface energy balance (e.g. Cullen et al., 2007; Oerlemans and Grisogono, 2002; Söderberg and Parmhed, 2006; van den Broeke, 1997; Smeets et al., 1998; Munro and Davies, 1978; Hoinkes, 1954; Kuhn, 1978; Munro and Scott, 1989).*

*Additional comment: Modelling the intermittency in stable boundary layers is very challenging and most numerical modelling studies, which are usually RANS model, do not capture these events. Our studies prove that LES are able to simulate such events, if the horizontal and vertical model resolution is sufficiently small to resolve most of the kinetic energy.*

*iii)*      *The scales are in the same order of magnitude as those found by other studies (e.g. Litt et al., 2015; Söderberg and Parmhed, 2006).*

> *Additional comment: Stable boundary layers show characteristic turbulence scales for the horizontal and vertical components. The simulation results show similar scales to those measured by other studies, which supports the choice of the grid resolution (that most TKE is resolved by the model) and the reliability of the subgrid-scale model (see later comment).*

*The comparison of the idealized LES simulations with the real-world observation is not possible. During the week of the 17th August 2014 we had temporarily installed two weather stations, one closed to Z1 on Zufallferner and another further down the valley. The glacier station measured between 13 and 14 h a mean wind velocity of 4.6 m/s at 2 m height above the surface. Even though this is closed to the simulated value (4.5-6 m/s, westerly flow), the two values are not comparable at all. The prescribed surface heating rate (1.2 K/h) of the surroundings is lower than the measured heating rate (4.1 K/hr) at that particular day. Furthermore, the idealized simulations do not account for differential heating by radiation which is important during the first two hours and leads to asymmetric cross-valley winds. Without doubt, the homogenous heating assumption is a major drawback of the code. Although the chosen heating rate is significantly lower and shadowing effects are absent the typical low level jet and the heat advection from the lateral boundaries are present. As indicated in the conclusion, due to conservative chosen boundary conditions the simulated advection effects might be weaker than the one observed in a real atmosphere.*

*We think there is a confusion why we use a surrogate atmosphere and what is the overall goal of this study. Therefore, we have updated the penultimate paragraph in the introduction as follows:*

*"To overcome this difficulty, we make use of high resolution Large-Eddy Simulations (LES). The LES are considered as pseudo-reality - a testbed to identify the shortcomings in the local surface heat flux estimates when the lack of observations restrict our micrometeorological knowledge to a few sites. The plausibility of the temperature interpolation algorithms and the derived surface heat fluxes can be more strictly tested in a surrogate world of atmospheric simulations, which offers a realization of atmospheric states in which all target variables are known. The pseudo-reality atmosphere is not required to be an observed real world case, but needs to be plausible realization of the atmosphere in the sense that relevant processes are realistically simulated. The advantage of such studies is that the surrogate atmosphere provides a perfect pseudo-observation of all the variables required to establish the skill of an interpolation method and hence the surface heat flux calculations. While surrogate atmospheres have been widely used in downscaling studies it's still a new approach in glaciological studies (Frias et al., 2006; Vrac et al., 2007; Maraun, 2012)."*

*We hope this emphasizes our overall goal to test the plausibility of interpolation algorithms and its consequences on the surface heat flux estimates. We neither make a statement that LES is superior to the bulk approach nor we claim it is a real case. It is simply a surrogate world of atmospheric states.*

**Comment 4)** There is confusion between point-site process understanding and interpolated/extrapolated input meteorological fields from sparse meteorological observation coming from on-glacier sites. If it's true and obvious that process understanding at individual sites is not sufficient to fully characterise the micrometeorological conditions over glacier surfaces, the practical or operational need to achieve such full characterization remain questionable (and in any case is not quantified in this paper).

> *AC: In fact, most scientific studies on glacier wide energy and mass balance are based on simple extrapolations of meteorological variables. Many of them use approaches based on linear gradients to create micrometeorological fields, while at the same time they report significant limitations in reproducing the spatial and temporal variability of glacier mass balance (e.g. MacDougall and Flowers, 2011; Gurgiser et al., 2013a; Prinz et al., 2016). This deficiency is also related to*

*shortcomings in the representation of sensible heat flux, since the sensible heat flux has proven to explain a great part of the melt energy and its variability at many glaciers (e.g. Braithwaite, 1995; Klok and Oerlemans, 2002; Gillett and Cullen, 2011; Conway and Cullen, 2013; Cullen and Conway, 2015). Based on the explanations presented above and the findings of the cited works, we think that there is no doubt about the scientific need of better and more realistic characterization of the micrometeorological conditions over glacier surfaces.*

**Comment 5**) Interpolation/extrapolation of meteorological data from on-glacier sites has limited practical usefulness. In operational model applications, there are almost no input data coming from inside the glaciers. In particular, I refer to applications aimed at exploring the climate sensitivity of glaciers, which is mentioned by the authors. Because the climatic sensitivity can be defined as 'the ratio of changes in the 2 m temperature above a glacier to changes in the temperature outside the thermal regime of that glacier (Greuell and Böhm, 1998), there is little usefulness in testing the errors coming from interpolation/extrapolation of pseudo-observed (or better, calculated) wind and temperature coming from points located inside the glaciers.

*AC: It is unclear what the referee means by "practical" and "operational". We however, try to address a well-defined research problem, namely the extrapolation of point observations of governing (micro-) meteorological parameters (temperature and wind) to a larger spatial scale.*

*There is a large number of studies focusing on glacier wide energy and mass balance modelling based on point observations. Some of them use on-glacier data (e.g. Hock and Holmgren, 2005; Sicart et al.,2005; Mölg et al., 2008, Reijmer and Hock, 2008; Mölg et al., 2009; MacDougall and Flowers, 2011; Sicart et al., 2011; Huintjes et al., 2015; Prinz et al. 2016), while others make recourse of off-glacier observations (e.g. Arnold et al., 1996; Klok and Oerlemans, 2002; Klok and Oerlemans, 2004; Gurgiser et al., 2013a; Gurgiser et al., 2013b)*

*The definition of "climatic sensitivity" as used by the referee and presented by Greuell and Böhm (1998), is not directly applicable to our research topic since we use the term "climate sensitivity" as an expression of a glaciers change in mass balance (rate) in response to a defined change in the ambient climate conditions (e.g. Oerlemans and Grisogono, 2002; Klok and Oerlemans, 2004; Mölg et al., 2008 and many more). The turbulent sensible heat flux plays a key role in process based analyses of climate change impacts to glaciers as it largely governs (together with longwave radiation) the sensitivity of a glacier to changes in air temperature (e.g. Braithwaite, 2009 and references presented above).*

*We agree that when the reaction of a glacier to changes in climate is examined, data from outside the (also changing) glacier boundary layer should be used (e.g. Klok and Oerlemans 2002). Nevertheless, this does not influence our conclusions since the main uncertainty potential in the calculation of micrometeorological fields highlighted by the current study is less determined by the origin of the observation data (on- or off-glacier), than by the applied extrapolation method. Consequently, our findings are not only valid for micrometeorological fields calculated from on-glacier stations, but for any kind of studies using linear gradients of temperature and wind to up-scale respective data.*

*However, our intention is to point out that the variations of the sensible heat flux in space and time cannot be sufficiently captured by simplified approaches. This may be negligible for glacier wide calculations of mass balance or melt during shorter time spans. But since especially mountain glaciers all over the world are currently undergoing rapid changes in shape/areal extent, the applied gradients might be not constant over longer time periods as changes in glacier extent influence the local microclimates. The same problem may appear under extraordinary conditions, such as for instance abnormal snow cover in the vicinity of the glacier, or years with changed mean synoptic flow, or other circumstances not reflected in the reference data set which was used to calculate the*

*gradients. It is hence obvious that more sophisticated methods are urgently needed to foster the understanding of the physical processes behind glacier changes.*

**Comment 6**) It could be more useful to test calculation schemes recently proposed in the literature (cited by the authors) starting from off-glacier weather stations.

*AC: We think the reviewer makes a good point and we have followed its recommendation and tested the Shea and Moore (2010) and the Greuell and Böhm (1998) temperature model. The Shea-model estimates the near-surface temperature on the glacier from the ambient temperatures using a piecewise linear regression approach. The model consists of our regression coefficients (T1, T$_*$, k1, k2), which need to be estimated for each station. The estimated coefficients are then related to morphometric measures, such as flow path length (FLP) and elevation. The estimation of the FLP is not trivial for unstructured grids and we had to develop our own code/algorithm to derive this measure. The code is based on a backtracking line search algorithm driven by local gradients. The attached Figure 1 shows the FLP estimates of the investigation area.*

*However, the lack of observations (only one observation at each station) makes it impossible to estimate the coefficients T1, T\*, k1 and k2 of the Shea-Model. We made some efforts using coefficient proposed by Shea and Moore (2010) and Carturan et al. (2015), but the model is very sensitive to the parameter choice. Additionally, relating these coefficients to the morphometric measures would introduce further seven coefficients. In our specific case, the problem is not well-posed at all and its impossible to calibrate the model.*

*Besides the Shea-model we also applied the temperature model proposed by Greuell and Böhm (1998). Basically, the model solves the change of heat within an air parcel travelling down an infinite slope. The approach requires the height of the katabatic wind H, the bulk transfer coefficient for heat $C_h$, FPL, a characteristic length, a location $x_0$ where the katabatic layer influences the air parcel, the mean slope, and the temperature T0 of the air parcel at x0. Most parameters can be calculated, except for $x_0$ which has to be determined. In literature different values for $x_0$ are given, ranging from 542 m. (Ayala et al. (2015)) to 1440 m (Carturan et al. (2015)). We have assumed a value of 1000 m for the analysis. The model was fitted to the observations by optimizing $C_h$. The model has been calibrated for each experiment using the same observations (S1 and S2) used for the linear interpolation. The estimated surface heat fluxes are 35.5 Wm-2 (westerly flow), 26.4 Wm-2 (easterly flow), 31.5 (northerly flow), and 28.8 Wm-2 (southerly flow).*

*We have updated Section 4.1 accordingly.*

Specific comments

**RC:** Page 1 line 5 and 7: please add the percentage in under-overestimations, and also the percent error in mass balance calculations. Small-scale heat flux, glacier heat fluxes... please be consistent throughout the paper and try to use always the same wording (i.e. sensible heat flux)

*AC: The focus of this study is the effect of local advection on the spatial sensible heat flux variation on glaciers and to test the skill of commonly used approaches to estimate the surface heat fluxes at a given point on the glacier and not the glacier mass balance. To do so the LES (including radiation) must be coupled directly with a distributed mass balance model and integrated over longer time periods. We are currently working on the coupling of the LES with a mass balance model.*

*We follow the recommendation of the reviewer and use the term sensible heat flux throughout the text.*

**RC:** Page 1 line 8 and 9: it is unclear if site selection and flow direction refer to data measurements, extrapolations, or validations

*AC: We have re-worded the sentence to clarify that the site selection refers to the extrapolated data: "The sign and magnitude of the differences depend on the site selection which are used for extrapolation as well as on the large-scale flow direction.".*

**RC:** Page 1 line 9-11: this is not adequately quantified in the paper. The magnitude of sensible heat flux calculation errors and their impact on the surface energy balance and on the derived climate sensitivities should be calculated and several numbers should be added also here in the abstract.

*AC: As discussed in the Comments 2,5 and the first specific comment, we do not quantify the impact of the errors in the sensible heat flux on the surface energy balance. We have decided to remove the last sentence from the abstract to avoid misunderstanding.*

**RC:** Page 2 line 10: consider replacing 'can make over' with 'can represent'

*AC: Done.*

**RC:** Page 2 line 13: consider removing 'peculiar'. In this period it is partly unclear to which mass balance studies dealing with small scale variations of melt rates the authors are referring to

*AC: The word 'peculiar' has been replaced by 'particular'.*

*The sentence has been re-phrased as follows: "Therefore, a profound knowledge of the advection processes and the micrometeorological characteristics is required to accurately calculate melt rates and their variations in space and time."*

**RC:** Page 2 line 16: what is meant exactly with 'the deficiency of monitoring activities'?

*AC: With 'deficiency' we refer to a falling short of a desirable number of observations. We think this is an unambiguous expression.*

**RC:** Page 2 line 26: an open scientific question

*AC: Done.*

**RC:** Page 2 line 27: be fully answered

*AC: Done.*

**RC:** Page 2 line 28: I suggest to state more clearly the aim(s) of the study

*AC: In the last two paragraphs of the introduction we now stress in more detail the aim of the paper (see also comment above):*

*"To overcome this difficulty, we make use of high resolution Large-Eddy Simulations (LES). The LES are considered as pseudo-reality - a testbed to identify the shortcomings in the local surface heat flux estimates when the lack of observations restrict our micrometeorological knowledge to a few sites. The plausibility of the temperature interpolation algorithms and the derived surface heat fluxes can be more strictly tested in a surrogate world of atmospheric simulations, which offer a realization of atmospheric states in which all target variables are known. The pseudo-reality atmosphere is not required to be an observed real world case, but needs to be plausible realization of the atmosphere in the sense that relevant processes are realistically simulated. The advantage of such studies is that the surrogate atmosphere provides a perfect pseudo-observation of all the variables required to establish the skill of an interpolation method and hence the surface heat flux calculations. While surrogate atmospheres have been widely used in downscaling studies it's still a new approach in glaciological studies (Frias et al., 2006; Vrac et al., 2007; Maraun, 2012)."*

**RC:** Page 3 line 19-20: SBL, SGS, please define acronyms

*AC: Done. SGS refers to subgrid-scale and SBL to stable boundary layer.*

**RC:** Page 4 line 9-10: it has been recognized

*AC: Changed.*

**RC:** Page 4 line 13-14: the negligibility of assumptions should be demonstrated and/or possible errors coming from assumptions should be quantified

*AC: We have added the following explanation to justify our assumption:*

*"Quantifying possible errors coming from this assumption or the model performance is challenging. The model can be tested either by a priori or a posteriori testing. The a priori test uses experimental or Direct Numerical Simulations (DNS) data to relate directly the residual-stress tensor given by the closure model. In an a posteriori test the accuracy of calculated statistics, such as mean wind or momentum flux, are compared with experimental data. Most LES approaches use a posteriori test to prove its applicability. Churchfield et al. (2014) has tested the Smagorinsky and bounded dynamic Langrangian model with the GABLS inter-comparison project (Global Energy and Water Cycle Experiment Atmospheric Boundary Layer Study, Beare et al., 2006) using a 6 m grid resolution. They found that both models are in line with the mean vertical profiles of wind speed, direction, potential temperature and variances. We therefore assume, that the backscatter of energy from the SGS model towards the resolved scales is negligible, if the LES resolves most of the turbulent kinetic energy (see Section 3.4)."*

*We have also added the following text to Section 3.4: "When decreasing the horizontal grid resolution to 25 m the resolved kinetic energy was only 60-70%. Additionally, a coarser grid leads to greater aspect ratios of the prismatic layers, which requires very short integration time steps (0.01 s) to guarantee stability. Increasing the prismatic layer heights is problematic since this affects the shear stress and momentum calculations closed to the surface. The choice of ~12.5 m is a good*

*tradeoff between computational costs and resolved scales."*

*And in Section 3.4:*

*"We have also tested the dynamic Smagorinsky model, but the simulations are found to be unstable due to large fluctuations of $C_s$."*

**RC:** Page 4 line 19-22 and page 6 line 14-16:  please see the previous comment on assumptions

*AC: It has been shown that the Lagrangian dynamic model, which averages $C_s$ over some volume backward in time along fluid particle paths, is appropriate for inhomogeneous flows (e.g. Pope, 2000; Anderson and Meneveau, 1999; Sarghini et al., 1999). It has been also successfully applied to the GABLS experiment (Churchfield, 2014), and complex terrain (Bou-Zeid et al., 2005). The references have been given in the text.*

**RC:** Page 6 line 23: I suggest adding some references for aerodynamic roughness heights

*AC: Done.*

**RC:** Page 6 line 25: ERA-Interim reanalysis data (also p5 l21)

*AC: Done.*

**RC:** Page 6 line 27-28: from which hour to which hour of the day?

*AC: The model has been initialized with the ERA-Interim profile from 06 UTC and a uniform surface temperature of 273.16 K (Section 2.3). Starting with the initial condition the model is integrated over a period of 9 hours. On p6L28 we refer to the last simulation hour. We have now added this information.*

*Please note that the idealized simulations do not account for differential heating by radiation (shading effect). The surface temperature of the surrounding is given by the prescribed surface heating rate (1.2 K/h). At the end of the simulation the surface temperature is 10.8 K.*

**RC:** Page 7 line 9-12: this part is somewhat unclear and it looks like the authors adjust the DEM (the only real-world component in this work) to the requirements of the numerical model. Is it correct? Please see comment to Page 4 line 13-14

*AC: Yes, we have relaxed the DEM to make use of period boundary conditions. Such boundaries require that faces on the opposite boundary (faces of grid cells) are equal within a certain tolerance. This is only possible, if the DEM grid points are equal on opposite boundaries. To do so the DEM grid points on opposite boundaries have been slowly displaced to match each other. The inner grid points are relaxed to get a smooth transition from the boundaries towards the inner domain. We have added a new figure showing a sketch of the relaxation procedure.*

**RC:** Page 9 line 4: please reword 'the intensity of the cross-valley circulation' to improve clarity

*AC: We have re-worded this part by: "the intensity of the slope winds".*

**RC:** Page 9 line 23-24: consider replacing 'do not jointly appear with high wind velocities' with something like 'do not appear in the areas with high wind velocities'

*AC: Done.*

**RC:** Page 10 line 34: can you quantify (or estimate) the percent contribution of the sensible heat flux to the total energy balance in your case study? This would be important for understanding the impact of calculated sensible heat flux on local-scale and area-averaged energy and mass balance

*AC: We agree that the contribution of the sensible heat flux to the total energy balance is important to understand the impacts on the mass-balance. However, the focus of this study is the effect of local advection on the spatial sensible heat flux variation on glaciers and to test the skill of commonly used approaches to estimate the surface heat fluxes at a given point on the glacier. In order to understand the impact, the LES (including radiation) must be coupled directly with a distributed mass balance model and integrated over longer time periods.*

**RC:** Page 11 line 10-11: can you provide some numbers in support to this statement?

*AC: The intermittency is a local and non-stationary process. The standard deviation of the vertical velocity fluctuations, $\sigma_w$, are low on the glacier (see Fig. 4) but are heavily right-skewed which indicates occasional mixing events. The power spectrum of the temperature signal shows that there are variations in the frequency of occurrence and amplitude of the mixing events. The scale-average time series show that there are average variances of up to $4.0\ C^2$. These burst events supply temporarily heat to the surface layer and the surface heat flux increases. This signal is neither present in the mean surface heat flux (Fig. 6) nor in the mean potential temperature (Fig. 7). We have now included references to the corresponding figures and chapters.*

**RC:** Section 3.4: in Figure 1 two weather stations are shown on the glaciers. Why data coming from these weather stations were not used for checking the reliability of LES experiments?

*AC: Please check back on comment 2, where we give an explanation why we can't compare the idealized LES with weather station data.*

**RC:** Page 11 line 19-23: with the authors, I recognize that this is a strong assumption, in particular over glaciers with such high range of elevation (2595-3750 m), quite different from the end-of-summer situation reported for Arolla by Brock et al., (2000). It should be possible to map the snow cover for the selected day, or to use another day with available snow cover data (e.g. from Landsat imagery). Alternatively, the authors should at least quantify the possible errors stemming from this assumption.

*AC: We know from the fields measurement during this period that there was a thin layer of fresh snow on the glacier. However, from that particular day there is no Landsat imagery available. Again, we like to remember these are idealized simulations and not real cases. Nevertheless, we have re-written the sentence as follows: "We assume similar roughness height for snow and ice since large parts of the glaciers were covered by a thin layer of fresh snow.".*

**RC:** Page 12 line 2-3: on which bases the authors say that the SGS model 'seems to work well' in their study?

*AC: In LES, the dynamics of the larger-scales are computed explicitly, while the smaller scales (residual stress tensor) are represented by the SGS model. Generally, the SGS model removes energy from the resolved scales to the residuals. In an a posteriori test we can test the accuracy of calculated statistics, such as mean wind or momentum flux, are compared with well know data from field experiments. We have shown that the calculated statistics, such as the integral turbulence scales, skewness and vertical velocity variance, are in the same order of magnitude as those obtained from observations. If the SGS model is too dissipative, the calculated measures would significantly differ from observations. In this pseudo-reality setup, it is difficult to prove the correctness of the SGS model. We have also tested the dynamic Smagorinsky model, but the simulations are found to be unstable due to large fluctuations of $C_s$.*

*We have added a new paragraph at the end of Section 3.4 to justify our conclusion:*

*"We have shown that the calculated statistics, such as the integral turbulence scales, skewness and vertical velocity variance, are in the same order of magnitude as those obtained from observations (e.g. Litt et al., 2015; Söderberg and Parmhed, 2006). If the SGS model is too dissipative, the calculated measures would be significantly lower than the observations. Which SGS model works best for stable boundary layers is not easy to tell, but the Lagrangian-averaged SGS model seems to work well in our study. We have also tested the dynamic Smagorinsky model, but the simulations are found to be unstable due to large fluctuations of Cs."*

**RC:** Page 12 line 4: maybe reword the title as 'Estimation of the sensible heat using the Bulk-Approach'

*AC: Done.*

**RC:** Page 12 line 20: replace 'given that' with 'in case' (I guess that it is meant where there is a weather station measuring the required variables)

*AC: Done.*

**RC:** Page 12 line 20-27: please consider moving this part in the following section

*AC: We followed the recommendation and moved this part in the following section.*

**RC:** Page 12 line 26-27: this is a strong statement, because there is complete absence of comparison between modelled and observed (relevant) processes. What are relevant processes? How can the authors assess that LES captures observations, without reporting observations or without citing literature on this topic?

*AC: Again we emphasize that the surrogate atmosphere can only be compared with well-known characteristics of stable boundary layers and dynamical atmospheric features obtained from in-situ measurements on alpine glacier (see comment 2). These characteristics include the vertical profiles (wind and temperature), sensible heat flux and turbulent structure of the boundary layer and should be of the same order of magnitude as the measurements. In Section 3.1, 3.2 and 3.3 we compare the wind magnitude, LLJ, intermittency and turbulence scales with observation made by other studies. We have shown that the calculated statistics, such as the integral turbulence scales, skewness and vertical velocity variance, are in the same order of magnitude as those obtained from observations (e.g. Litt et al., 2015; Söderberg and Parmhed, 2006).*

*We have re-written this sentence as follows:*

*"As demonstrated in Sec. 3.1, 3.2 and 3.3, the LES provide plausible vertical wind and temperature profiles, surface heat fluxes and turbulent structures of the boundary layer."*

**RC:** Page 13 line 3: please replace 'pseudo-observed' with 'calculated'. I guess these are temperature and wind speed data calculated using LES, is it right? Please specify

*AC: The sentence now reads as:*

*"The simulated (LES) wind velocities and temperatures at the two sites were linearly extrapolated across the glacier (e.g. Paul and Kotlarski, 2010; Machguth et al., 2009; Huintjes et al., 2015; Weidemann et al., 2013; Jarosch et al., 2012)."*

**RC:** Page 13 line 5: surface heat flux, surface sensible flux, or surface sensible heat flux? Please be consistent

*AC: We have check the manuscript for consistency.*

**RC:** Page 13 line 7-9: please explain why there are differences at the two Za and Z0 sites, given that (in my understanding) wind speed and temperature at these sites are the same using the bulk method and the LES (i.e. they differ in the rest of the analysed area, but not at Za and Z0).

*AC: This is correct. The differences at the locations should be zero, but there are small differences of about 1-3 $Wm^{-2}$. The sensible heat flux from the LES is calculated online at each time step. The fluctuating $L_s$ and U can significantly increase the calculated sensible heat flux over a short time period. For the bulk approach we used the mean $L_s$ and U calculated over the last hour, which could cause small differences in the calculations. If there is only a small difference from zero the station is attributed to one of the contour classes and it might seem that there is a large difference between the LES and the bulk method at the two sites.*

**RC:** Page 13 line 20: the average sensible heat flux (please, add % error in the text). How big is the impact on glacier-wide total energy balance calculations?

*AC: The sentence now reads as:*

*"On a glacier-scale, the bulk approach underestimates the average heat flux between 5.2 (-16.6%) and 6.9 W m−2 (-20.3%) for the westerly, easterly and northerly flow (see Tab. 2). The local differences for the southerly case, however, almost cancel each other out (0.8 W m−2, 2.2%)."*

*We have not computed the glacier-wide total energy balance and therefore cannot estimate the impact of the differences on the energy balance (see comment above).*

**RC:** Page 13 line 27: for which wind direction?
**RC:** Page 13 line 28: using linear extrapolations across the glaciers?

*AC: We now explicitly mention the wind direction:*

*"In the following, we estimate the sensible heat flux according to Eq. 17 for each combination of S1 and S2-S5 using the linearly extrapolated temperature and mean wind velocity (the mean value of the two sites) from the westerly flow case."*

**RC:** Page 13 line 29: in my opinion there is an equivocal use of the term 'bulk method', which is a method for calculating turbulent exchanges, referred to the calculations using linear extrapolations across the glaciers. I would suggest clarify/avoid ambiguities

*AC: Yes, we agree that the term 'bulk estimate' is not used correctly. We have re-worded the sentence as follows:*

*"Fig. 11 shows the differences between the sensible heat fluxes calculated by the bulk approach and the surrogate atmospheres."*

**RC:** Page 13 line 33 and in the following: please check or clarify, if gradients are too large (in absolute value) underestimations of temperature and sensible heat flux should occur in the upper parts of the glaciers. Moreover, in absence of model validation, why the LES model has to be the right one and the Bulk has to be the wrong one, a priori?

*AC: Yes, this statement is wrong. The correct statement is:*

*"In the case that stations are located in a region of strong temperature advection (e.g. case S1-S2 and S1-S3) the derived temperature gradient is too shallow, and the bulk approach overestimates the sensible heat fluxes in most regions of the glacier. Similarly, temperature gradients are too steep when stations are protected from warm air transport, and on average fluxes are underestimated (e.g case S1-S4 and S1-S5)."*

*As initially mentioned (comment 3), we neither make a statement that LES is superior to the bulk approach nor we claim it is a real case. It is simply a surrogate world of atmospheric states. The advantage of such studies is that the surrogate atmosphere provides a perfect pseudo-observation of all the variables required to establish the skill of an interpolation method and hence the surface heat flux calculations.*

**RC:** Page 14 line 3: also in this case I suggest to calculate the relative importance of these errors in the overall energy balance of the glacier

*AC: Please see our response to the first specific comment.*

**RC:** Page 14 line 5-8: this part is methodological and should be moved at the beginning of Sect. 4.2. It also deserves rephrasing to improve clarity

*AC: The part has been re-written and moved to the beginning of Sec. 4.2*

*"This approach has two major implications: i) the temperature field is completely decoupled from the flow and therefore disregards local flow features (e.g. gap flows and bluff bodies), and ii) the wind velocities are too low over large areas on the glacier."*

**RC:** Page 14 line 12: it is unclear why the authors selected only a clear-sky case study

*AC: In order to draw a general conclusion, however, a large number of experiments is needed to cover the wide spectrum of topographic and atmospheric constellations. Unfortunately, we have already reached our computational capacities and try to solve this in an upcoming project. Each LES run of 9 hours' simulation time requires a computational time of 5-7 days on a High-Performance Computer with 400 cores. For that reason, we have focused on a clear-sky case of which we have expected pronounced thermal wind phenomena and heat advection.*

**RC:** Page 14 line 17-22: I have several points, which could/should be at least partly addressed or discussed in the manuscript. In particular they concern: i) the practical or operational need to fully characterise the micrometeorological conditions over glacier surfaces; ii) the linear extrapolation of forcing fields from sites placed over glaciers (again, almost never available in practical model applications); iii) related to the previous point, the climate sensitivity has to be assessed with respect to climatic conditions
observed outside the microclimatic influence of the glaciers.

  i)   *This issue has been addressed in Comment 4*
  ii)  *This issue has been addressed in Comment 5*
  iii) *This issue has been addressed in Comment 5*

**RC:** Page 14 line 20: here and elsewhere, I suggest speaking about differences and not errors, because the comparison is between calculations and not between calculations and observations

*AC: Yes, we agree and have changed the wording.*

**RC:** Page 14 line 30: percent error of what?

*AC: The number gives the mean error by which the bulk approach differs from the actual LES values. We have added this information to the number.*

**RC:** Page 15 line 1-2: when small-scale variations of surface energy balance are required? Please add this in the introduction and recall it here and/or in the abstract

*AC: The phrase now reads as: "We can conclude that a profound knowledge of the heat advection process is needed when small-scale variations of surface energy balance are required for distributed mass balance studies."*
*The need for small-scale variation of the sensible heat flux for accurate melt rates and their variation in space and time has been made more clear in the introduction.*

**RC:** Page 14 line 6: using off-glacier stations for what?

*AC: We have removed this part of the sentence.*

Comments on the figures and tables:

**RC:** Figure 1: this image lacks east-north coordinates or inset displaying wider geographical setting of the study area. Four weather stations are reported, whose data are not used in this paper

*AC: Done.*

**RC:** Figure 2 (and following maps): I suggest adding some contour line (or hillshaded DTM, like in Fig. 1), which is needed for a better understanding of the local topography and of its effects on the calculated variables

*AC: Actually, the figures do have a hillshading, but the glacier surface is very smooth so that it is not particularly eye-catching. We have made an attempt to add some contour lines to the figures, but the figures appear overloaded due to its small size. We would prefer to keep the figures as they are and think that Figure 1 gives all the essential information.*

**RC:** Figure 9: in the caption just begin with 'differences in the surface....' and correct pseudo-observations coherently with the text

*AC: Done.*

**RC:** Table 3: I suggest adding LES estimates and % differences (not error, please correct also in Table 2) as in Table 2.

*AC: The bulk estimates are always compared to the same LES case (westerly flow, 33.8 $Wm^{-2}$). Therefore, we think it is not necessary to include the LES estimate in Table 3. However, the % differences have been added to Table 3.*

**References**

*Arnold, N.S., Willis, I.C., Sharp, M.J., Richards, K.S., and Lawson, W.J.: A distributed surface energy-balance model for a small valley glacier. I. Development and testing for Haut Glacier d' Arolla, Valais, Switzerland, Journal of Glaciology, 42, 140, 77-89, 1996.*

Braithwaite, R.J.: Aerodynamic stability and turbulent sensible-heat flux over a melting ice surface, the Greenland ice sheet, Journal of Glaciology, 41, 139, 562-571, 1995

Braithwaite, R. J.: Calculation of sensible-heat flux over a melting ice surface using simple climate data and daily measurements of ablation, Annals of Glaciology, 50, 9-15, 2009.

Conway, J.P., and Cullen N.J.: Constraining turbulent heat flux parameterization over a temperate maritime glacier in New Zealand, Annals of Glaciology, 54, 41–51, 2013.

Cullen, N. J. and Conway, J. P.: A 22 month record of surface meteorology and energy balance from the ablation zone of Brewster Glacier, New Zealand, Journal of Glaciology, 61, 931–946, 2015.

Gillett, S. and Cullen, N. J.: Atmospheric controls on summer ablation over Brewster Glacier, New Zealand, International Journal of Climatology, 31, 2033–2048, 2011.

Gurgiser, W., Mölg, T., Nicholson, L., and Kaser, G.: Mass-balance model parameter transferability on a tropical glacier, J. Glaciol., 59, 845–858, 2013a.

Gurgiser, W., Marzeion, B., Nicholson, L. I., Ortner, M.; and Kaser, G.: Modeling energy and mass balance of Shallap Glacier, Peru, The Cryosphere, 7, 1787-1802, 2013b.

Hock, R. and Holmgren, B.: A distributed surface energy-balance model for complex topography and its application to Storglaciären, Sweden, J. Glaciol., 51, 25–36, 2005.

Huintjes, E., Sauter, T., Schröter, B., Maussion, F., Yang, W., Kropácek, J., Buchroithner, M., Scherer, D., Kang, S., and Schneider, C.: Evaluation of a coupled snow and energy balance model for Zhadang glacier, Tibetan Plateau, using glaciological measurements and time-lapse photography, Arctic, Antarctic, and Alpine Research, 47, 573–590, 2015.

Klok, E.J., and Oerlemans, J.: Model study of the spatial distribution of the energy and mass balance of Morteratschgletscher, Switzerland, Journal of Glaciology, 48, 505-518, 2002.

Klok, E.J., and Oerlemans, J.: Modelled climate sensitivity of the mass balance of Morteratschgletscher and its dependence on albedo parameterization, International Journal of Climatology, 24, 231-245, 2004.

MacDougall, A.H., and Flowers, G.E.: Spatial and Temporal Transferability of a Distributed Energy-Balance Glacier Melt Model, Journal of Climate, 24, 1480-1498, 2011.

Mölg, T., Cullen, N.J. Hardy, D.R. Kaser, G., and Klok, L.: Mass balance of a slope glacier on Kilimanjaro and its sensitivity to climate, International Journal of Climatology, 28, 881-892, 2008.

Mölg, T., Cullen, N.J., Hardy, D.R., Winkler, M., and Kaser G.: Quantifying climate change in the tropical mid-troposphere over East Africa from glacier shrinkage on Kilimanjaro, Journal of Climate, 22, 4162-4181, 2009.

Oerlemans, J., and Klok, E. J.: Energy Balance of a Glacier Surface: Analysis of Automatic Weather Station Data from the Morteratschgletscher, Switzerland, Arctic, Antarctic, and Alpine Research, 34, 4, 477-485, 2002.

Oerlemans, J., and Grisogono, B.: Glacier winds and parameterisation of the related surface heat fluxes, Tellus A, 54, 440–452, 2002.

Oerlemans, J.: The Microclimate of Valley Glaciers. Utrecht Publishing and Archiving Services, Universiteitbibliotheek Utrecht, 138 pp, 2010.

Prinz, R., Nicholson, L. I., Mölg, T., Gurgiser, W., and Kaser, G.: Climatic controls and climate proxy potential of Lewis Glacier, Mt. Kenya, The Cryosphere, 10, 133-148, 2016.

Reijmer, C., and Hock, R.: Internal accumulation on Storglaciaren, Sweden, in a multi-layer snow model coupled to a distributed energy-and mass balance model, Journal of Glaciology, 54, 61-72, 2008.

Senese, A., Diolaiuti, G., Mihalcea, C., and Smiraglia, C.: Energy and Mass Balance of Forni Glacier (Stelvio National Park, Italian Alps) from a Four-Year Meteorological Data Record. Arctic, Antarctic, and Alpine Research: February 2012, Vol. 44, No. 1,122-134, 2012.

Sicart, J.E., Wagnon, P., and Ribstein, P.: Atmospheric controls of the heat balance of Zongo Glacier (16°S, Bolivia), Journal of Geophysical Research, 110, 1-17, 2005.

Sicart, J.E., Hock, R., Ribstein, P., Litt, M., and Ramirez, E.: Analysis of seasonal variations in mass balance and meltwater discharge of the tropical Zongo Glacier by application of a distributed energy balance model, Journal of Geophysical Research, 116, 1-18, 2011.

Smeets, C.J.P.P., Duynkerke, P.G., and Vugts, H.F.: Turbulence Characteristics of the Stable Boundary Layer Over a Mid-Latitude Glacier. Part I: A Combination of Katabatic and Large-Scale Forcing Boundary-Layer Meteorology, 87, 117-145, 1998.

Frías, M. D., E. Zorita, J. Fernández, and C. Rodríguez-Puebla (2006), Testing statistical downscaling methods in simulated climates, Geophys. Res. Lett., 33, L19807, doi:10.1029/2006GL027453.

Vrac, M., M. L. Stein, K. Hayhoe, and X.-Z. Liang (2007), A general method for validating statistical downscaling methods under future climate change, Geophys. Res. Lett., 34, L18701.

Maraun, D. (2012), Nonstationarities of regional climate model biases in European seasonal mean temperature and precipitation sums, Geophys. Res. Lett., 39, L06706, doi:10.1029/2012GL051210.